# Dynamically assembled photochromic cages operational in water with visible light

Valentin Schäfer [1], Angelika Seliwjorstow [1], Olaf Fuhr[2,3] & Zbigniew L. Pianowski [1,4,5]

Producing chemical nanostructures that can mimic the efficient adaptability of complicated biological systems to environment changes is among the main goals of nanotechnology. Progress in this area requires understanding of the adaptation mechanisms towards external stimuli at the molecular level. Due to rapid and precise spatiotemporal addressability, light-driven dynamic systems are particularly attractive for such mechanistic studies. Here, we show efficient formation of dynamic covalent cages that undergo a series of reversible constitutional changes driven by visible light. Their complex, yet predictable and often quantitative response to irradiation and other external stimuli (metal ions, pH) reveals design principles that can be applied to assemble adaptable molecular machines showing life-like behavior. Upon reduction of the dynamic imine bonds, stable covalent cages are isolated. Their response to red light, also in aqueous media, indicates the potential for in vivo applicability, as red light can deeply penetrate human tissues.

Supramolecular self-assembly of functional building blocks enables preparation of diverse nanostructures, such as cages or macrocycles, with a broad range of functions directly related to the nanoconfinement−like allosteric receptors[1], regulated catalysis[2], or modulation of photophysical properties of guest molecules[3]. The nanostructures can respond to external stimuli, such as pH, temperature, or chemical input, in order to change their configuration or selectivity towards guest molecules[4]. Particularly advantageous are nanostructures resulting from dynamic covalent self-assembly, as they show self-sorting behavior in response to external stimuli or reagents−the initial complex mixtures of intermediates can often be directed into well-defined heteroleptic or homoleptic species[5]. Light is a highly advantageous stimulus to be applied in such systems, as it does not permanently contaminate the reaction, and can be used with high spatiotemporal precision. The commonly used antenna to convert the light energy into effects at the molecular level are molecular photoswitches[6], such as azobenzenes[7], indigoids[8], DASA, norbornadienes, dihydropyrenes, or hemipiperazines[9–13]. A variety of molecular photoswitches has been incorporated into nanocages and macrocycles[14,15], yielding photoresponsive nanostructures, including diarylethenes[16], sterically crowded alkenes, diazocines[17], and azobenzenes. Yet, most of the demonstrated nanostructures were operational in organic solvents, but not in water−due to lack of solubility, hydrolysis, or incompatibility of the switching mechanism. Moreover, switches triggered with red and near-infrared light are highly demanded, as these wavelengths (> 630 nm, therapeutic window of light) can efficiently penetrate soft human tissues rich in hemoglobin, and thus have broad applicability potential in human photopharmacology[18–21] or as in vivo receptors[22].

In 2021, we demonstrated[23] a tetra-*ortho*-fluoroazobenzene bis-aldehyde **1** as a photoswitch triggered with red light (660 nm) and a useful intermediate in soft material preparation. This compound has been used by the Schmidt group in an elegant work to dynamically assemble macrocycles that change their constitution upon irradiation[24,25]. Around that time, the Feringa group demonstrated UV-light-driven (340 nm) destabilization of a dynamically assembled cage[26]−the stable (E)-configured cage based on a non-halogenated azobenzene bis-aldehyde resists an imine exchange, while the more

[1]Institute of Organic Chemistry IOC, Karlsruhe Institute of Technology, Karlsruhe, Germany. [2]Institute of Nanotechnology INT, Karlsruhe Institute of Technology, Karlsruhe, Germany. [3]Karlsruhe Nano Micro Facility KNMFi, Karlsruhe Institute of Technology, Karlsruhe, Germany. [4]Institute of Biological and Chemical Systems – Functional Molecular Systems IBCS-FMS, Karlsruhe Institute of Technology, Karlsruhe, Germany. [5]International Institute of Molecular Mechanisms and Machines, Polish Academy of Sciences IMol PAN, ul. Flisa 6, Warszawa, Poland. e-mail: pianowski@kit.edu

strained (Z)-configured cage can be decomposed using an external reagent. This process can be inverted upon violet light-induced (420 nm) (Z)-to-(E) back-isomerization followed by thermal equilibration, showing the dynamic nature of imine-based photochromic cages. Interestingly, only the *meta*-configured azobenzene architecture produced well-defined cages, while the *para*- and *ortho*-isomers did not form cage-shaped species. That work was further expanded to reversible cage-ring phototransformations[27]. Other interesting systems based on photoreactive cages bearing azobenzene functions have been recently reported by the groups of Beves[28], Greenaway[29], Kataev[30], Wang[31], and Clever[17].

## Results and discussion

We were intrigued whether the compound **1** can be used for assembly −via dynamic covalent chemistry−of well-defined supramolecular cages sensitive to red light, as well as responsive to other external stimuli. We were also curious whether modifications of the switch structure (replacing fluorine with chlorine atoms) may yield a chromophore switchable with higher wavelengths (such as near-infrared light), and still capable of forming photochromic cages. Other interesting opportunities were the formation of heteroleptic species and the implementation of other triggers that may shift the equilibrium in such dynamic systems, such as transition metal ions. Upon demonstrating all that, we fixed the dynamic outcome of our assembly by irreversible formation of stable covalent bonding via imine reduction. The resulting stable cage can be reversibly transferred between the organic and aqueous phase upon protonation−even $CO_2$ is sufficiently acidic−and can be reversibly photoswitched in water with visible light frequencies.

### Formation of a stable cage via dynamic covalent chemistry

We mixed (in the 3:2 ratio) the photochromic bis-aldehyde **1** (**FA**) in the thermodynamically stable (E)-configuration with the *tris*(2-aminoethyl)-amine **2** (**EN**)−a flexible trivalent linker that enables dynamic imine bond formation with aldehyde substrates−in chloroform. After 24 h incubation, we isolated a well-defined dynamic covalent cage **3** (**FA₃EN₂**) with 60% yield (Fig. 1a, Supplementary Fig. 4A). Its stability was sufficient to grow crystals and characterize **3** via X-ray diffraction (Fig. 2a). If the ratio of **1** was slightly decreased, we observed−in addition to **3**−the incompletely formed cage **4** (**FA₂EN₂**), with one missing **FA** pillar. Yet, **3** resolubilized in chloroform after purification (filtration, re-dissolving, and trituration with ethyl acetate) did not produce **4** nor other intermediates. It demonstrates that **3** is the most stable form in that system. The isolated (E,E,E)-**3** cage has been almost quantitatively (> 95%) photoisomerized to the (Z,Z,Z)-**3** cage with red light (660 nm). We believe that our system shows a pronounced templating effect, as we did not observe any meaningful amounts of the mixed (E,Z,Z) or (E,E,Z) forms at 660 nm. Incorporation of the switch into the cage geometry slightly reduced its thermal stability (from 312 h to 123 h at 25 °C in CDCl₃, Supplementary Figs. 7 and 8) in comparison with the free aldehyde **1**. The cage has shown well-pronounced photochromism in solution (Fig. 1b). The forms (E,E,Z)-**3** and (E,Z,Z)-**3** could be observed as minor products upon irradiation with green light (523 nm), but their lifetime is much shorter (< 7 h) than the (Z,Z,Z)-**3** (Supplementary Fig. 6, 9–10). Interestingly, we could also form the (Z,Z,Z)-**3** cage with good yields upon incubation of the bis-aldehyde **1** pre-irradiated with red light (660 nm, containing >85% of the (Z)-**1** form under these conditions) with the triamine **2** upon 24 h (Supplementary Figs. 4B, 11, and 12).

### Reduction of 3 yields a stable cage 5, reversibly switchable in water

The (E,E,E)-configured dynamic cage **3** was treated with NaBH₄ and produced a stable covalent cage (E,E,E)-**5**, where the imine bonds have been reduced to secondary amines. Both species formed crystals and

their structures could be determined via X-ray diffraction (Fig. 2a). The (E,E,E)-**5** can be photoisomerized with amber (590 nm) or green (523 nm) light, resulting in almost quantitative formation of the (Z,Z,Z)-**5**. This isomer upon exposure to violet light (430–410 nm) is fully converted. The majority (>70%) of it forms the (E,E,E)-isomer. The remaining 30% is distributed between the (E,E,Z)- and (E,Z,Z)-form (Supplementary Fig. 13 and 14). Due to the diminished π-electron system, **5** is not sensitive to 660 nm light anymore (and with 620 nm, it produced a mixture of isomers). While **5** is insoluble in water, protonation with TFA quantitatively transfers it into the aqueous phase (Fig. 2b). The resulting **5**\*H⁺ cage can be reversibly switched between two states. One containing predominantly (> 60%) (Z,Z,Z)-isomer (590–523 nm) and the other >70% (E,E,E)-isomer (410 nm; Supplementary Table 2 and Supplementary Figs. 16–21). Even carbonic acid is strong enough to fully transfer **5** from chloroform to the aqueous phase saturated with $CO_2$. Upon stirring overnight in an open vial, $CO_2$ merely evaporates, and **5** fully returns to chloroform (Fig. 2c). We have also investigated the formation of inclusion complexes between the cage **5**\*H⁺ and cucurbiturils. Based on the diameter of **5** obtained by X-ray diffraction analysis (-8 Å, estimated using Mercury©), we chose cucurbit[8]uril (**CB8**) as host (inner diameter of the cavity = 8.8 Å), and we observed spectral shifts which may attest to supramolecular interactions, such as inclusion complex formation (Supplementary Figs. 22 and 23)[30]. The complexes were irradiated with 523 nm (green light) and 410 nm (violet light), and then treated with 10 eq. excess of amantadine (protonated 1-aminoadamantane), which shows high affinity (> 10¹²M⁻¹) to **CB8**, and thus completely removes the cage from the complex (Supplementary Figs. 24–27 and Supplementary Table 3). We have noticed that irradiation of **5** inside the inclusion complex increases the preference for the forms containing (Z)-configured isomers, as compared with photoisomerization in free aqueous solution (Supplementary Fig. 26 and Supplementary Table 3).

To further assess the biocompatibility of our system operational in water with visible light wavelengths, we have performed a colorimetric assay for cell metabolic activity in the presence of our cage (so-called MTT assay used as standard for cytotoxicity determination). The (E,E,E)-**5**\*H⁺ isomer showed the value of $IC_{50} = 0.58\,\mu M$, while the mixture of isomers obtained at the photodynamic equilibrium with irradiation of 523 nm light (with the majority of (Z,Z,Z)-isomer) showed a slightly higher value of $IC_{50} = 1\,\mu M$ (Supplementary Fig. 30 and Supplementary Table 4). While the general toxicity at the sub-micromolar level can be explained, e.g., by the physical, detergent-like interactions of the cationic species with the anionic phospholipid components of cellular membranes, its photomodulation (less than an order of magnitude) seems too little to be useful in any practical application. Here, we can also see that the cage **5**\*H⁺ can be safely used in biological setups at concentrations below 0.5 μM.

### Dynamic covalent chemistry of the cage 3 with monovalent reagents

Next, we examined the stability of cage **3**. We treated the stable isomer (E,E,E)-**3** with 3 equiv. of benzaldehyde or benzylamine−two monovalent reagents competing for the imine bond. After 7 days of incubation at 20 °C, no significant exchange has been observed. Yet, upon generation of the (Z,Z,Z)-**3** isomer and its incubation with these reagents over the same time period, we have seen significant exchange with benzaldehyde, and almost quantitative with benzylamine (Supplementary Figs. 31–33). It demonstrates that the (Z,Z,Z)-configured cage **3** is less stable than the (E,E,E) isomer, and that benzylamine is more reactive in the replacement process than benzaldehyde.

In contrary, the free aldehyde (E)-**1** reacts with benzylamine **18** quantitatively, forming the (E)-bis-imine (E)-**17** (Supplementary Fig. 35), which upon irradiation with red light (660 nm) yields 89% of the (Z)-**17** (Supplementary Fig. 36)− photoconversion similar to the bis-aldehyde **1**, but lower than for the **3**. The t₁/₂ of the (Z)-**17** was determined to be

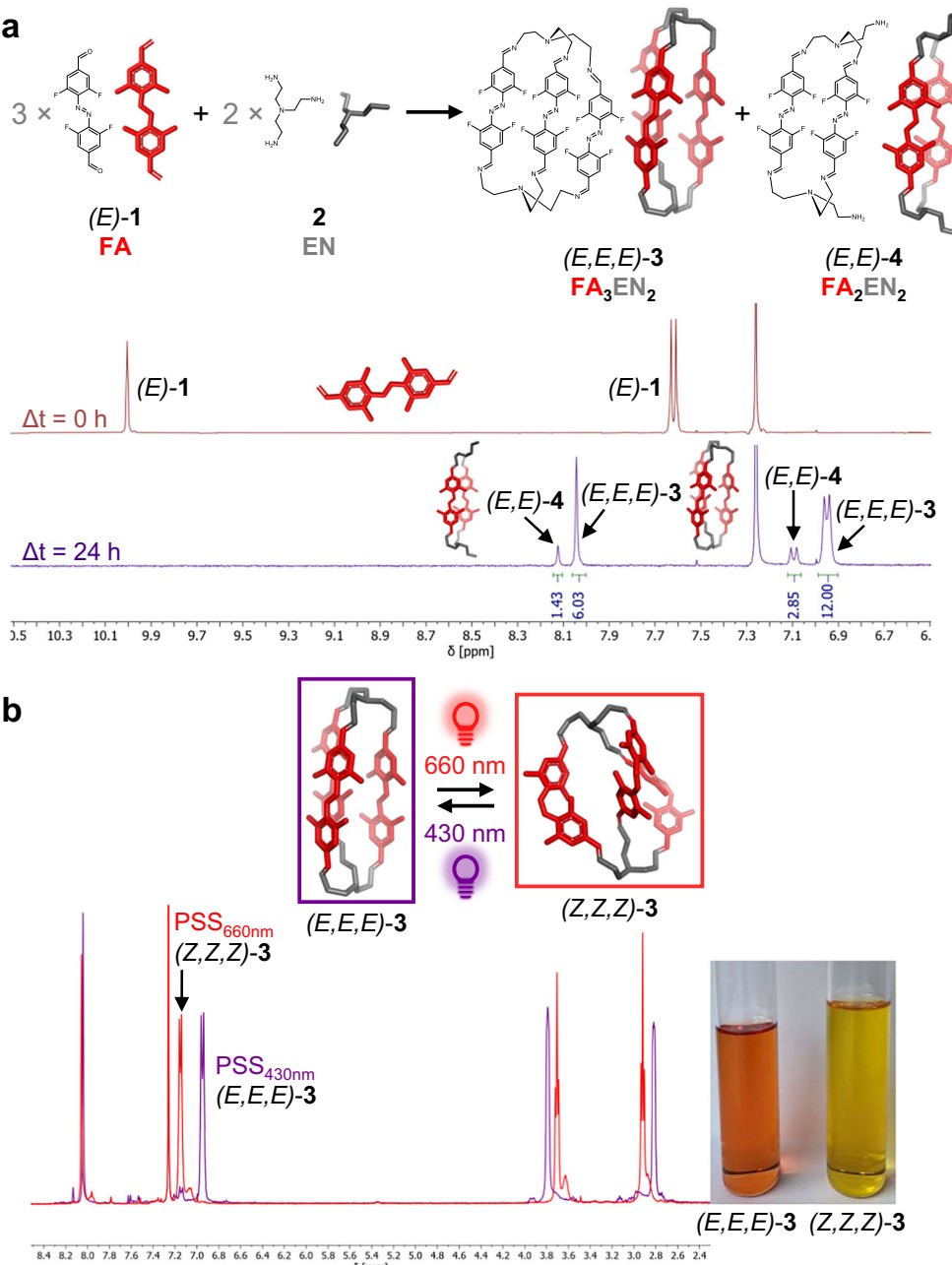

**Fig. 1 | Dynamic formation and photochromic properties of cage 3. a** Formation of the cage *(E,E,E)*-**3** in CDCl₃ within 24 h; (**b**) photochromism of the cage **3**—the orange *(E,E,E)*-isomer is almost quantitatively (> 95%) converted to the yellow *(Z,Z,Z)*-**3** with red light (660 nm; $t_{irr}$ = 45 min) and recovered (> 90%) with violet light (430 nm; $t_{irr}$ = 15 min); ¹H NMR spectrometer frequency = 400 MHz.

239 h (25 °C in CDCl₃) (Supplementary Fig. 37)—slightly reduced in comparison to the bis-aldehyde *(Z)*-**1** ($t_{1/2}$ = 312 h), but still significantly higher than for the *(Z,Z,Z)*-**3** cage ($t_{1/2}$ = 123 h). Thus, both the enhanced photoconversion and the decrease in thermal stability for the photochromic cage seem to derive mainly from the discussed template effect in the cage. This is corroborated with the fact that during thermal back-isomerization of *(Z,Z,Z)*-**3**, no *(E,Z,Z)*- or *(E,E,Z)*-isomers were detected, but only the *(E,E,E)*-**3** product.

### Exchange of cage 3 with non-halogenated photochromic bis-aldehyde

Then, we exposed cage **3** to a bis-aldehyde **6** containing the non-halogenated azobenzene chromophore (**AZ**). Again, the *(E,E,E)*-**3** did not undergo significant reaction upon at least 15 days of incubation.

Yet, the *(Z,Z,Z)*-**3** obtained upon exposure of the sample to red light (660 nm) demonstrated a significant exchange. According to NMR spectra, multiple species have been generated at this stage (Supplementary Fig. 40). But upon back-isomerization of the mixture with 430 nm light (which restores the *(E)*-configuration on both fluorinated and nonfluorinated azobenzenes (see Table 1 and Supplementary Table 6), we have isolated only the unchanged cage *(E,E,E)*-**3** and a compound *(E,E,E)*-**7** together with the fluorinated bis-aldehyde *(E)*-**1**. The compound **7** occurred to be a cage analog of **3**, where one of the three azobenzene pillars has been exchanged from the tetra-fluorinated to the nonfluorinated block (Fig. 3 and Supplementary Fig. 41).

We did not isolate any cage products of further exchange in measurable quantities (although respective ESI traces could be

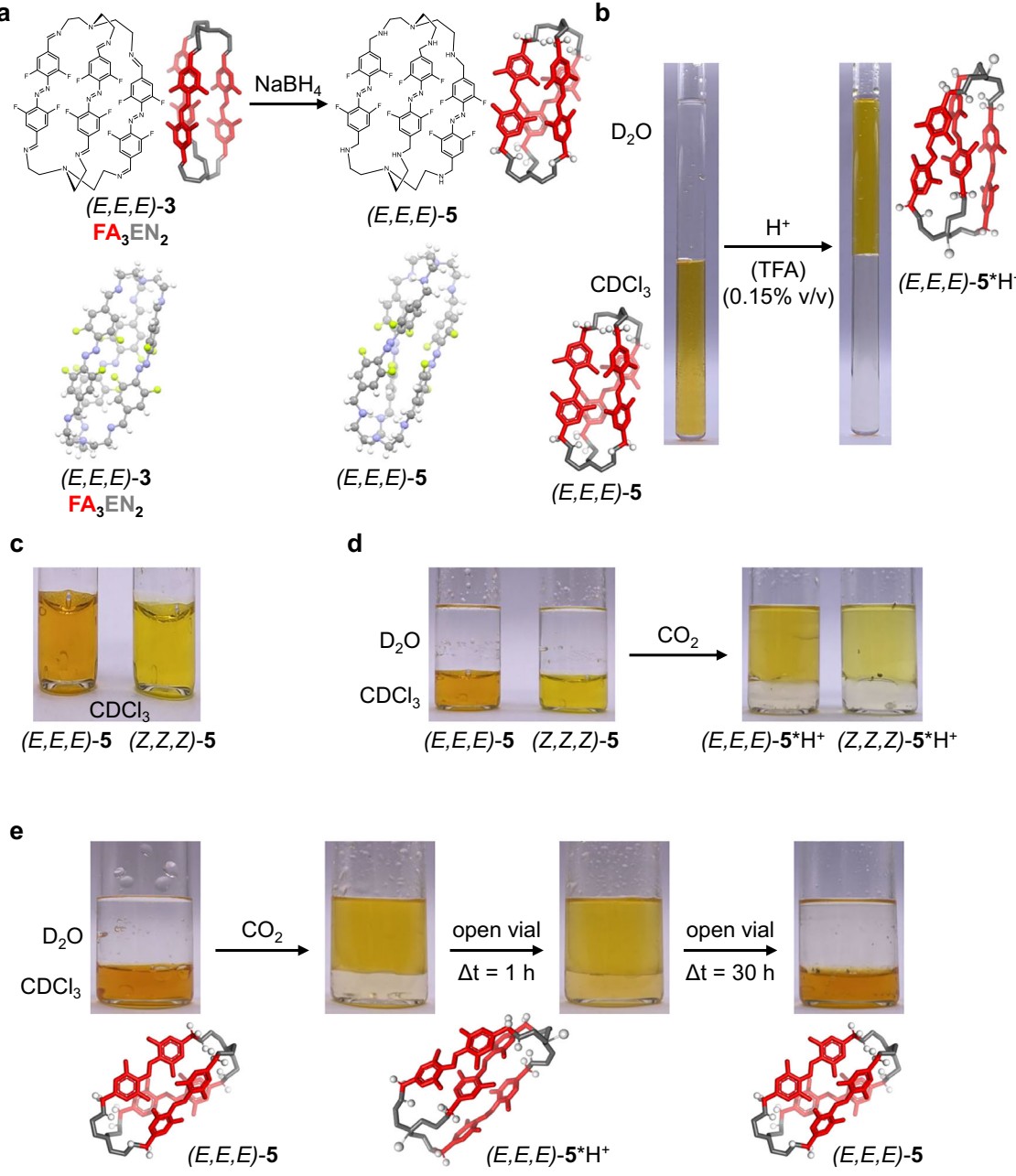

**Fig. 2 | Reduction of the dynamic cage 3 to stable covalent cage 5 and the behavior of 5 upon protonation. a** Structure of the dynamically assembled cage *(E,E,E)*-**3** can be analyzed with X-ray crystallography. It can be reduced with NaBH₄ which converts the labile imine bonds to stable secondary amines. The structure of resulting covalent cage **5** can be imaged with X-ray analysis as well; (**b**) extraction of **5** from the CDCl₃ phase (c = 1 mM) into the D₂O phase, facilitated by the addition of TFA (0.15% v/v, 20 mM); (**c**) *(E,E,E)*−**5** and *(Z,Z,Z)*−**5** dissolved in CDCl₃ (1 mM); the

*(Z,Z,Z)*-isomer was obtained by 523 nm irradiation (tᵢᵣᵣ = 45 min); (**d**) left: *(E,E,E)*−**5** and *(Z,Z,Z)*−**5** dissolved in CDCl₃ (1 mM, lower layer); right: the same samples after bubbling CO₂ through the solution for 10 min; (**e**) **5** was dissolved in CDCl₃ (1 mM) and then mixed with D₂O, resulting in no extraction into the water phase. A constant stream of CO₂ bubbled through the solution for 10 min was sufficient to transfer **5** into the water phase. Stirring for 30 h in the open vial reversed the process, bringing **5** back into the organic layer.

registered, Supplementary Fig. 45), also upon elongated periods of incubation (Supplementary Fig. 46). This result corroborates the report of the Feringa group[26] where the *para*-configured *bis*-aldehyde **6** does not form well-defined cages with the tris-amine **2**, but forms precipitation, likely containing linear polymers. The structure of cage **7** has been confirmed by DOSY (Supplementary Fig. 41B). We have also reduced the reaction mixture containing the **7** with NaBH₄ and identified the mass of the corresponding reduced heteroleptic cage **8** (Supplementary Fig. 49).

## Tetra-*ortho*-chlorinated azobenzene bis-aldehyde addressable with near-IR light−its photochromic properties and dynamic exchange with the cage 3

The bis-aldehyde **1** can be triggered with red light due to the expansion of its π-electron system in comparison with the unmodified tetra-*ortho*-fluoroazobenzene chromophore (which is inert to >600 nm light)[23]. We were interested, if the extension of π−electron system in a tetra-*ortho*-chlorinated chromophore (already addressable with red light[18]) will expand the range of its light response even further. To

check that, we have synthesized the chlorinated bis-aldehyde **9** (**CA**) (Fig. 4a) and characterized its photochromism with various wavelengths of light. While the fluorinated aldehyde **1** photoisomerizes from *(E)*- to *(Z)*-form with green or red light (523–660 nm), the chlorinated **9** is photoconverted with red light (623 or 660 nm) from the thermally stable *(E)*-isomer to the mixture containing the majority of the *(Z)*-form. Yet, the reverse photoconversion (to mixtures containing >75% of the *(E)*-**9**) occurs either upon green/blue light irradiation (455–523 nm) or upon exposure to the near-IR light (730 nm) with additional cut-off filter for wavelengths below 715 nm, (see Table 1). This observation makes **9** a rare switch that can be bidirectionally

addressable with wavelengths within the therapeutic window of light (650–900 nm). That property might be used in the future for precise temporary in vivo photocontrol of photopharmacological systems, or photomodulation in complex biological setups.

We used **9** to generate another cage system. Upon mixing of **9** with the triamine **2** over 20 hours at elevated temperature (65 °C), we observed complete substrate consumption and formation of two new species (Fig. 4b), later identified as a fully formed cage **10** with three chlorinated azobenzene pillars and a partially opened cage **11** (ca. 15% of the isolated product, see Supplementary Fig. 54)—the assembly where three chlorinated **CA** pillars are bound with the two triamines **EN** with total 5 imine bonds (leaving one aldehyde and one amine group unreacted). The ESI-MS identification is supported with DOSY NMR spectra that show similar hydrodynamic radii of both species (Supplementary Fig. 55). This result indicates that formation of the complete cage **10** is more sterically demanding than in the case of fluorinated cage **3**, due to increased steric hindrance and changed geometry of the **CA** pillars.

The cage *(E,E,E)*-**10** can be isolated upon repetitive trituration with ethyl acetate. It was irradiated with red light (660 nm), to cause the *(E)* → *(Z)* photoisomerization of the chlorinated azobenzene fragments. The resulting *(Z,Z,Z)*-**10** cage is likely unstable—due to even larger steric demands of the *(Z)*-tetrachloroazobenzene-containing systems in comparison to the respective *(E)*-isomers. Thus, multiple species are produced. Yet, irradiation of that mixture with light in the range 523–430 nm (Supplementary Fig. 57) quickly restores the original distribution of the cage *(E,E,E)*-**10** and the minor component *(E,E,E)*-**11** (collectively denoted as **Mix-*E*10**), proving the highly dynamic properties of our system.

We performed five full red/blue switching cycles (660 nm/455 nm), which restored the original **Mix-*E*10** product distribution at the end (see Supplementary Figs. 58 and 89). Then, we wanted to use NIR light (730 nm) or thermal equilibration at room temperature to reconstitute the cage *(E,E,E)*-**10** previously dissipated with red light. However, this was not successful—most of the soluble fractions

**Table 1 | Photoconversions of the tetra-*ortho*-chloroazobenzene bis-aldehyde 9 and its thermal half-life in comparison with the fluorinated analog 1; *cut-off filter below 715 nm; (E), (Z) – geometric isomers of the N = N-bond ((E)-opposite substituents, (Z)-substituents on one side)**

| λ(irradiation) (nm) | 9 (CA) | | 1 (FA) | |
|---|---|---|---|---|
| | % (E)-isomer | % (Z)-isomer | % (E)-isomer | % (Z)-isomer |
| Dark state | >95 | 0 | >95 | 0 |
| 730* | 75 | 25 | – | – |
| 700 | 66 | 34 | – | – |
| 660 | 32 | 68 | 11 | 89 |
| 623 | 38 | 62 | 10 | 90 |
| 590 | 53 | 47 | 11 | 89 |
| 523 | 80 | 20 | 30 | 70 |
| 490 | 88 | 12 | 56 | 44 |
| 455 | 93 | 7 | 81 | 19 |
| 430 | 93 | 7 | 91 | 9 |
| Thermal half-life (t₁/₂) at 25 °C | 4.1 h | | 312 h | |

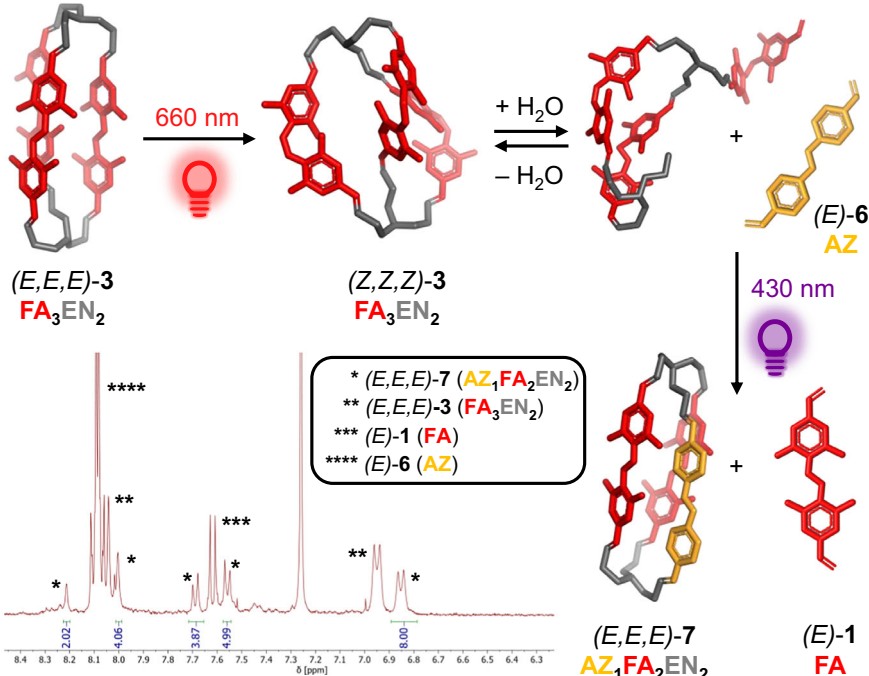

**Fig. 3 | Exchange of the pillars in the cage 3 exposed on the nonfluorinated azobenzene bis-aldehyde 6.** The ¹H NMR spectrum shows the reaction mixture after 15 days (see also Supplementary Fig. 44). The only newly formed product **7** (along with the expelled fluorinated bis-aldehyde **1**) is a heteroleptic cage with one non-halogenated and two halogenated azobenzene pillars. Further exchange products have not been found. The mixture was kept at PSS₆₆₀ₙₘ by repeated short irradiation intervals (20 min) during the experiment; ¹H NMR spectrometer frequency = 400 MHz.

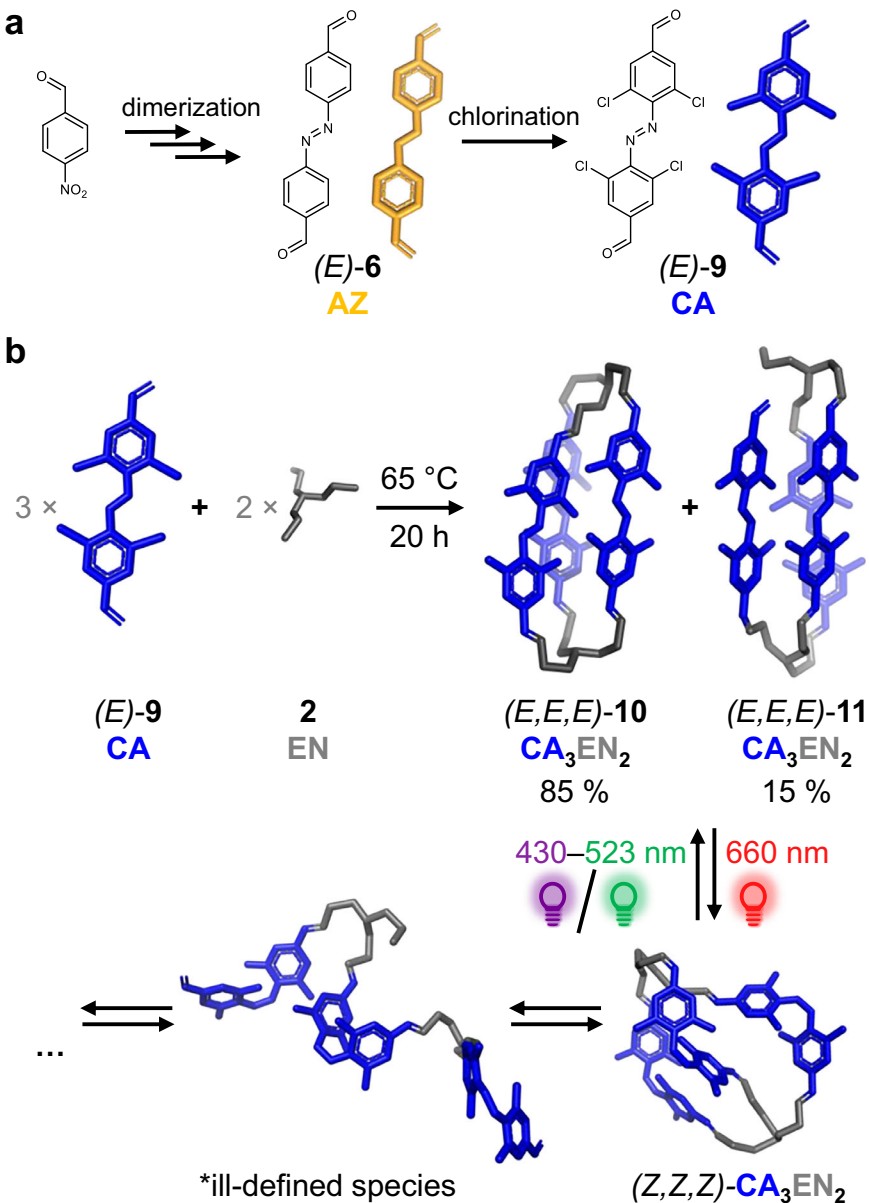

**Fig. 4 | Formation of chlorinated cage 10 and its dynamic behavior upon irradiation. a** The photochromic chlorinated *bis*-aldehyde **9** is synthesized from the non-halogenated azobenzene *bis*-aldehyde **6** in a single step; (**b**) the aldehyde **9** reacts with the tris-amine TREN **2** and forms with good yields (76% overall), the closed cage **10** along with the partially opened cage **11** (in 85:15 ratio); red-light irradiation results in reversible disassembly of the cage structure.

contained the bis-aldehyde **9**, while the rest of the material was converted into insoluble linear polymerization products (likely because the cage reformation was too slow due to steric hindrance, Supplementary Fig. 59). Yet, if the thermal equilibration was performed at 50 °C, the original **Mix-E10** distribution was fully restored within 4 hours (Supplementary Fig. 60). In another experiment, we partially destabilized the **Mix-E10** with NIR light (730 nm). Upon irradiation of the resulting composition with blue light (455 nm), we recovered the **Mix-E10** (Supplementary Fig. 61) upon two switching cycles. In that experiment, the disruption of the cage is less pronounced—in line with the lower ratio of (*Z*)-**9** produced at 730 nm (25% at the $PSS_{730nm}$) than at 660 nm (68%, see Table 1).

To further explore the dynamic nature of our system, we incubated the pre-formed fluorinated cage **3** with the chlorinated pillar **9**. When performed with the (*E*)-isomers in the dark, very slow conversion (<10%) was observed within 1 week, eventually equilibrating after

1 month, with ~20% of a new species, which was identified as the heteroleptic cage **12** (with one chlorinated and two fluorinated pillars) (Fig. 5 and Supplementary Fig. 62). If the reaction is initiated by irradiating the sample with 660 nm (>95% (*Z,Z,Z*)-**3** and >60% (*Z*)-**9**), immediate formation of several new species was observed (Supplementary Fig. 63) and after 1 week, >30% of **12** formed. Yet, thermal equilibration (1 month) of that system in darkness resulted in partial reestablishment of the cage **3** after (*E*)-configuration of both azobenzene pillars is thermally restored, resulting in the same final component ratio as in the process in the absence of light.

**Covalent cage addressable with red light in water**
Encouraged by the efficient formation and green-light-induced switching of the covalently fixed cage **5** based on fluorinated azobenzene, described above, we attempted to prepare an analogous cage with the chlorinated bis-aldehyde **9**. For that, we tried to reduce

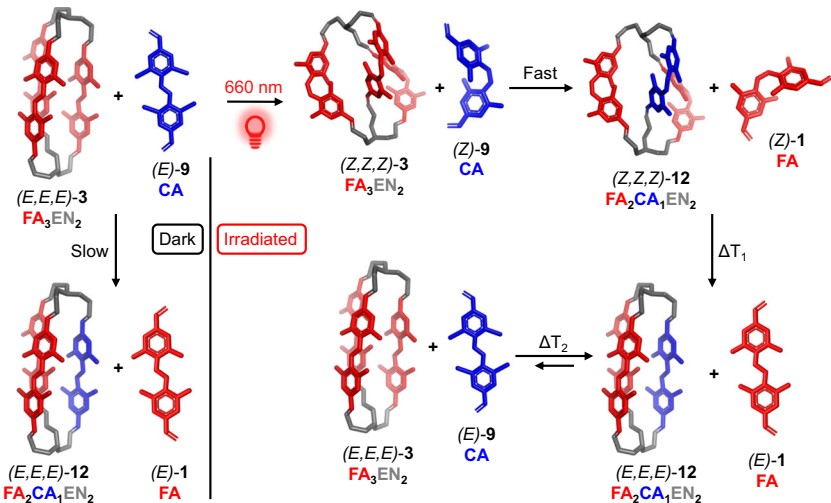

**Fig. 5 | Generation of the heteroleptic cage 12 from the fluorinated cage 3 and the chlorinated bis-aldehyde 9.** The cage **12** can be assembled from the irradiated or non-irradiated cage **3** via dynamic exchange, yet with different rates and yields.

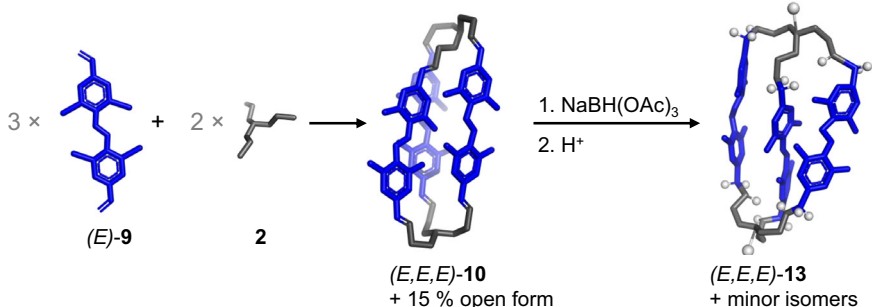

**Fig. 6 | Synthesis of the covalent cage 13 switchable in water with red light (660 nm).** The cage **10** is dynamically assembled from **9** and reduced using NaBH(OAc)$_3$, yielding the stable cage **13**.

the dynamic cage system **Mix-*E*10**. The attempts with NaBH$_4$ or BH$_3$*THF were unsuccessful, yielding an inseparable mixture of many substances. Yet, reduction with NaBH(OAc)$_3$ showed sufficient selectivity to isolate the desired covalent cage **13** in fair yield (15% over two steps from the bis-aldehyde *(E)*-**9** and TREN **2**; Fig. 6 and Supplementary Figs. 64 and 65).

The cage **13** (which contains predominantly one isomer in the dark state, which we assume to be the *(E,E,E)*-**13**) can be also protonated with TFA, and in this state shows good solubility in water. We have irradiated the protonated *(E,E,E)*-**13**\*H$^+$ in aqueous solution with 660 nm (red light within the therapeutic window range), and the *(E,E,E)*-isomer was fully converted to the mixture of photoisomers. Green light irradiation (523 nm) produced isomer mixture with different component distribution, while the blue light (455 nm) restored the original isomer distribution− predominantly the *(E,E,E)*-**13** isomer− observed in the non-irradiated sample (NMR experiments, see the Supplementary Fig. 68). These results were corroborated with UV−Vis spectra of the dark and irradiated states (Supplementary Fig. 66), where the absorbance of the dark sample is almost fully restored upon 455 nm irradiation. Due to the insufficient amount of the isolated cage, we could not perform the cytotoxicity tests.

### Exchange of the cage 3 with a non-photochromic bipyridine bis-aldehyde and in situ formation of a multivalent ligand for transition metals

In the last section, we demonstrate that our dynamic cage can react with metal ions as a stimulus orthogonal to the effects of irradiation

described above. By the principles analogous to the aforementioned cage **3**, the triamine **2** (**EN**) can form a non-photoresponsive homoleptic cage **15** with 3,3′-bipyridine-4,4′-bis-aldehyde **14**. The process is slow and incomplete in organic solvents, but the well-defined product has been identified with an approximate yield of 40% (judged by $^1$H NMR and MS) (Supplementary Figs. 69−70 and 71A). For further characterization, we were also able to isolate **15** on a larger scale by precipitation (supporting information).

As the cage **15** seems to be less stable than the photochromic cage **3**, there is only partial interconversion observed, if **3** is mixed with the respective bipyridine pillar **14** in chloroform (Supplementary Fig. 72). Both the expected heteroleptic cages−the product of single (**16a**) and double (**16b**) exchange−are formed as minor products (Fig. 7a and Supplementary Figs. 71B and 73). Interestingly, while the cage **16a** with two azobenzene pillars can be dissipated with red light and restored with blue light irradiation, the cage **16b** (one remaining azobenzene) is inert to both light frequencies, probably due to the steric congestion (Supplementary Fig. 71C).

Upon formation of the homoleptic cage **15** from **2** and **14**, we added water to the reaction mixture in chloroform. Under these conditions, none of the resulting components is extracted into aqueous phase (Supplementary Fig. 75). Yet, if we replace water with a solution of transition metal salts, like Fe, Co, Ni, or Cu, the cage **15** forms multivalent colorful complexes with the metal ions, which are instantly and quantitatively transferred into the aqueous phase (only the unreacted pyridine bis-aldehyde **14** used in excess remains in the organic phase, as it does not form complexes with the used metal

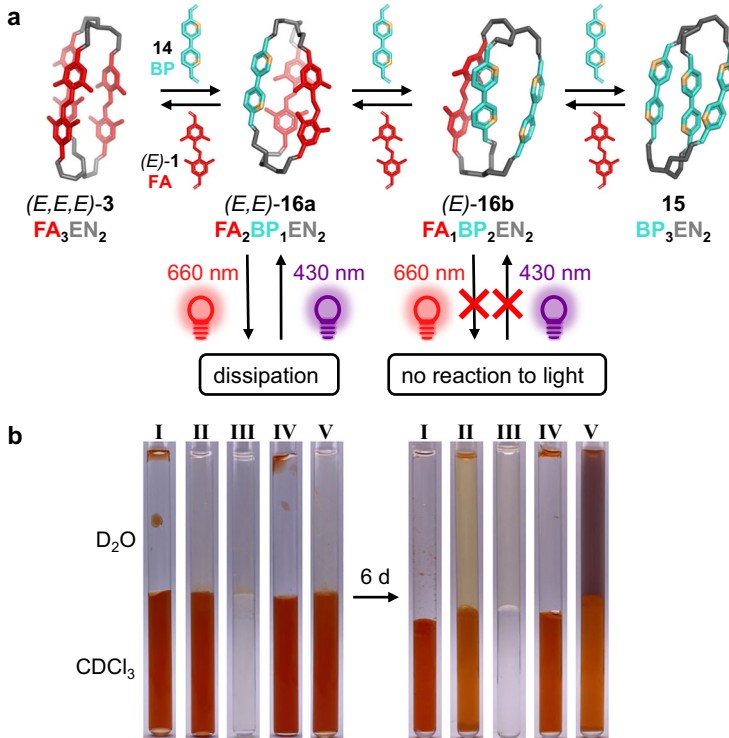

**Fig. 7 | Dynamic exchange process between the cage 3 and the non-photochromic bis(pyridilaldehyde) 14 results in different outcome in the presence or in the absence of metal ions (Fe²⁺). a** The non-photochromic cage **15** can be formed upon mixing **EN** with the bis(pyridilaldehyde) **14**, or (partially) upon exchange of **3** with **14** (together with heteroleptic cages **16a** and **16b**). The cage **16a** can be dissipated with 660 nm red light, and reconstituted with 430 nm light; (**b**) visualization of the exchange experiments between the cage **3** and the pyridine bis-aldehyde **14**, and the respective Fe²⁺-mediated transfer from CDCl₃ (lower phase) into D₂O (upper phase); **I**: CDCl₃: **3** (2 mM) / D₂O: −; **II**: CDCl₃: **3** (2 mM) / D₂O: FeCl₂ (20 mM); **III**: CDCl₃: **14** (6 mM) / D₂O: FeCl₂ (20 mM); **IV**: CDCl₃: **3** (2 mM) + **14** (6 mM) / D₂O: −; **V**: CDCl₃: **3** (2 mM) + **14** (6 mM) / D₂O: FeCl₂ (20 mM).

ions), (Fig. 7b and Supplementary Fig. 76A). The efficient formation of the cage **15** in presence of metal ions[32–34], as well as similar molecular cages[33,34], has been already described in literature, whereby a 1:2 binding ratio between **15** and metal ions like Zn²⁺ and Fe²⁺ was reported. Following these results, we determined the association constants $K_{a1}$(**15*Fe²⁺**) = 806 (±13) M⁻¹ and $K_{a2}$(**15*2Fe²⁺**) = 1789 (±67) M⁻¹ (Supplementary Figs. 125–127).

Also if we mix the pre-formed cage (*E,E,E*)-**3** with the bipyridine bis-aldehyde **14** in presence of aqueous iron(II) chloride solution, the heteroleptic bipyridine-containing cages **16a** and **16b** are quantitatively transferred into the aqueous phase with concomitant complexation of the iron ion, leaving only residual amounts of the cage **3** and the photochromic fluorinated bis-aldehyde **1** as the exchange by-product (Fig. 7b, sample **V**).

Prior research has demonstrated numerous photochromic dynamic covalent assemblies with diverse topologies and assembly efficiency (Fig. 8 and Table 2). We have shown that halogenated azobenzene-derived bis-aldehydes can efficiently form cages, which undergo dynamic and reversible exchange in response to various external stimuli (other bis-aldehydes, metal ions) in a light-dependent manner, upon exposure to visible light frequencies. The dynamic outcome can be fixed upon reducing the dynamic imine bonds to permanent covalent amine bonds. The resulting stable cages can be bidirectionally and reversibly switched with a pair of colors in the visible light range—green/violet for the fluorinated system, and red/blue for the chlorinated one—in organic solvents, as well as upon protonation in water. The fluorinated stable cage works as a simple $CO_2$ sensor within a biphasic setup. We also reported that one of our further cage components—the chlorinated azobenzene bis-aldehyde—shows bidirectional addressability with light frequencies within the therapeutic window

(660 nm (*E*) → (*Z*), 730 nm (*Z*) → (*E*)), and thus is a promising chromophore for in vivo applications, including photopharmacology. We demonstrated that our dynamic cages undergo formation and reversible exchange with high efficiency and a predictable outcome. We believe, that especially our cages operational in aqueous media with red light make a step forward toward the construction of biocompatible molecular machines—also via self-assembly, which will be applicable for cargo storage, or stimuli-dependent selective catalysis.

## Methods

### Synthesis of the cage (*E,E,E*)-3

The fluorinated *bis*-aldehyde (*E*)-**1** (150 mg, 484 μmol, 1.00 eq.) was dissolved in anhydrous chloroform (220 mL) under an argon atmosphere. To the solution, the triamine (TREN) **2** (47.2 mg, 48.3 μl, 323 μmol, 0.667 eq.) in 5 mL chloroform was added dropwise. The resulting mixture was heated to 62 °C for 24 h. Afterwards, the solution was cooled down to 20 °C, filtered to remove insoluble material, and the filtrate was dried *in vacuo*. The resulting red powder was purified by trituration with ethyl acetate (5 × 5 mL), redissolved in chloroform, filtered again, and dried under reduced pressure. A final washing of the solid with ethyl acetate (5 mL) yielded the cage (*E,E,E*)-**3** (99.0 mg, 88.8 μmol, 55%) as a dark red solid.

### Synthesis of the cage (*E,E,E*)−5 upon reduction of the dynamic cage (*E,E,E*)−3

To a solution of (*E,E,E*)-**3** (60.0 mg, 53.8 μmol, 1.00 eq.) in chloroform (11 mL) and methanol (11 mL) was added sodium borohydride (NaBH₄, 42.8 mg, 1.13 mmol, 21.0 eq.) under argon counter flow. The mixture was stirred for 24 h at 20 °C, during which the color of the solution changed from red to orange. Afterwards, the solvents were removed

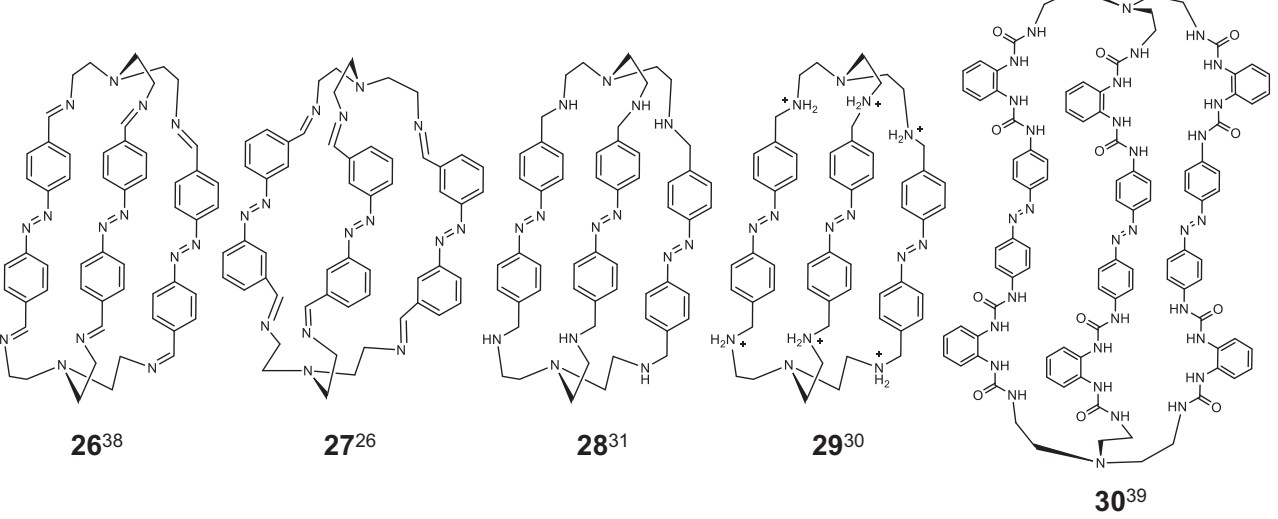

**Fig. 8 | Photochromic molecular cages, previously reported in the literature, structurally related to cages 3 and 5.** Their fundamental parameters are summarized in Table 2.

## Table 2 | Summary of fundamental parameters of the compounds presented in this work, followed by structurally related cages from literature for comparison (structures shown in Fig. 8)

| Compound | $\lambda_{max,abs}$ (E) or (E,E,E) (nm) $\pi$–$\pi^*$ \| n–$\pi^*$ | (Z)- or (Z,Z,Z)-ratio (%) \| @$\lambda_{irr}$ (nm) | (E)- or (E,E,E)-ratio (%) \| @$\lambda_{irr}$ (nm) | $t_{1/2}$ (h) @25 °C (Z) → (E) or (Z,Z,Z)→(E,E,E) |
|---|---|---|---|---|
| **1** | 314 \| 476 | >85 \| 660 | >90 \| 430 | 312 |
| **17** | n.d. | >85 \| 660 | n.d. | 239 307[a] |
| **3** | 322 \| 483 | >95 \| 660 | >90 \| 430 | 123 321[a] |
| **5** | 311 \| 459 | ~90 \| 590 | >60 \| 430 | n.d. |
| **5*H⁺** [b] | 310 \| 455 | >70 \| 590 | >70 \| 410 | n.d. |
| **9** | 286 \| 459 | ~25 \| 730 >65 \| 660 | >85 \| 490 >90 \| 455 | 4.1 |
| **10** | 287 \| 459 | n.d. \| 730 quant. (E,E,E)-conversion \| 660 | >95 \| 455 | <4 h @50 °C[f] |
| **26**[38] | n.d. | degradation \| 365 | n.d. | n.d. |
| **27**[26c] | 322 \| 436 | 68 \| 340 | 85 \| 420 | <10 h @60 °C[f] |
| **28**[31] | 325 \| 440 | 87 \| 365 | 85 \| 420 | n.d. |
| **29**[30d] | 320 \| 430 | traces \| 365 10 \| 365 (NaF) | 74 \| ambient light | 8.5 |
| **30**[39e] | 390 \| 470 | 94 \| 380 | >99 \| 520 | 2.7 |

[a]DCM-d₂; [b]H₂O/D₂O; [c]benzene/benzene-d₆; [d]D₂O acetate buffer pH 3.6; [e]DMSO/DMSO-d₆; [f]quantitative (E,E,E)-reformation; n.d. = no data.

Unless noted otherwise, the data were acquired in CHCl₃/CDCl₃; (E), (Z) – geometric isomers of the N = N-bond ((E)-opposite substituents, (Z)-substituents on one side).

under reduced pressure. The resulting solids were redissolved in a biphasic mixture of chloroform (15 mL) and saturated aq. NaHCO₃ solution (15 mL) and vigorously stirred for 30 min, whereupon a clear orange organic layer and a turbid aqueous layer formed. The layers were separated and the aq. layer was extracted with chloroform (3 × 15 mL). The solvent of the combined organic layers was removed under reduced pressure, yielding the cage *(E,E,E)*−**5** (53.4 μmol, 60.2 mg, 99%) as orange powder.

### Synthesis of the cage (E,E,E)-10

The chlorinated *bis*-aldehyde *(E)*-**9** (40.0 mg, 106 μmol, 1.00 eq.) was dissolved in anhydrous chloroform (55 mL) under an argon atmosphere. To the solution, the triamine (TREN) **2** (10.4 mg, 10.6 μl, 71.0 μmol, 0.667 eq.) in 5 mL of chloroform was added dropwise. The resulting mixture was heated to 62 °C for 12 h. Afterward, the solution was cooled down to 20 °C, filtered to remove insoluble material, and the filtrate was dried *in vacuo*. The resulting brown powder was purified by trituration with ethyl acetate (5 × 5 mL), redissolved in chloroform, filtered again and dried under reduced pressure. A final washing of the solid with ethyl acetate (5 mL) yielded the cage *(E,E,E)*-**10** (35 mg, 26.7 μmol, 76%) as a brown powder.

### Synthesis of the cage (E,E,E)-13 upon reduction of the dynamic cage (E,E,E)-10

The chlorinated *bis*-aldehyde *(E)*-**9** (30.0 mg, 79.8 μmol, 1.00 eq.) was dissolved in 40 mL of anhydrous CHCl₃ under an argon atmosphere. N',N'-Bis(2-aminoethyl)ethane-1,2-diamine (7.78 mg, 7.97 μL, 53.2 μmol, 0.667 eq.) was dissolved in 2 mL of anhydrous CHCl₃ and added dropwise over the course of 1 h. The resulting mixture was stirred at 23 °C for 7 days. Then, the solution was filtered (0.2 μm), cooled to 0 °C, and solid NaBH(OAc)₃ (118 mg, 559 μmol, 21.0 eq.) was added under argon counter flow. The mixture was slowly warmed up to 23 °C and stirred for 20 h. Afterwards, it was again cooled to 0 °C and 3 mL of MeOH were added slowly. After stirring for another 2 h, 15 mL H₂O were added, followed by 50 mL of saturated NaHCO₃ solution. The aqueous layer was extracted with 50 mL of CHCl₃, the combined organic layers were washed with H₂O (2 × 30 mL), dried over Na₂SO₄, and the solvent was evaporated under reduced pressure. The crude product was dissolved in a MeOH/MeCN/TFA mixture (70/20/10) and purified using preparative HPLC (MeCN/H₂O, 0.1% TFA; gradient = MeCN 20% → 70% over 60 min), yielding **13** (9 mg, 15%, overall yield from cage components) as a dark orange solid.

### Solvents and reagents

Solvents (per analysis quality) were commercially purchased (Acros Organics, Fisher Scientific, Sigma-Aldrich, Eurisotop, VWR) and used without further purification if not stated otherwise. Reagents were commercially purchased from industry companies (ABCR, Acros, Alfa Aesar, Fisher Scientific, Merck, BLDpharm, VWR) and used without further purification if not stated otherwise.

## Graphs and spectra

Graphs were plotted and fitted in OriginPro, Version 2023 (Copyright © 1991–2022 OriginLab Corporation, Northampton, MA, USA). NMR spectra were processed using MestReNova, Version 14.1.2 (Copyright © 2005–2020 Mestrelab Research S.L., Santiago de Compostela, Spain).

## Graphics

Chemical structures, molecular graphics and crystal structures were created and visualized using ChemDraw (Version 20.1, Copyright © 1985–2020 PerkinElmer Informatics, Inc., Waltham, MA, USA), the PyMOL Molecular Graphics System (Version 3.1.4, Copyright © Schrödinger, LLC, New York, NY, USA), and Mercury (Version 3.6, Copyright © 2001–2015 Cambridge Crystallographic Data Centre, Cambridge, UK), respectively.

## Analytical balance

Sartorius Basic (PreZion: 0.001 g), Sartorius M2P Micro Balance (0.001 mg) and Mettler Toledo AE163 (0.01 mg) were used.

## Nuclear magnetic resonance spectroscopy

$^1$H-NMR and 2D-spectra were recorded using Bruker AM 400 (400 MHz) and Bruker Avance DRX 500 (500 MHz). $^{13}$C-NMR spectra were recorded using Bruker AM 400 (101 MHz) and Bruker Avance DRX 500 (126 MHz). $^{19}$F-NMR spectra were recorded using Bruker AM 400 (376 MHz) with a $^1$H–$^{19}$F composite pulse decoupling sequence. All measurements were carried out at 25 °C. The spectra were analyzed according to the first order. All coupling constants are absolute values and expressed in Hertz (Hz). The following solvents from Eurisotop, Thermo Fisher Scientific, and Sigma-Aldrich were used: chloroform-d$_1$, dichloromethane-d$_2$, acetonitrile-d$_3$, trifluoroacetic acid-d$_1$ and D$_2$O. Chemical shifts δ were expressed in parts per million (ppm) and referenced to chloroform-d$_1$ ($^1$H NMR: δ = 7.26 ppm, $^{13}$C NMR: δ = 77.16 ppm), dichloromethane-d$_2$ ($^1$H NMR: δ = 5.32 ppm), acetonitrile-d$_3$ ($^1$H NMR: δ = 1.94 ppm), D$_2$O ($^1$H NMR: δ = 4.79 ppm). The signal structure is described as follows: s = singlet, d = doublet, t = triplet, q = quartet, p = quintet, b = broad, m = multiplet.

## Infrared spectroscopy

IR spectra were recorded on a Bruker IFS 88 using ATR (attenuated total reflection) for solids. The absolute intensities of the peaks are given as follows: vs = very strong 0–9%, s = strong 10–39%, m = medium 40–69%, w = weak 70–89%, vw = very weak 90–100%.

## Mass spectrometry

The mass spectra were measured on a Finnigan MAT 95 (70 eV) for EI-MS (electron ionization mass spectrometry) or FAB-MS (fast atom bombardment mass spectrometry) (with 3-nitrobenzyl alcohol as matrix). ESI-MS (electron spray ionization mass spectrometry) was conducted on a Thermo Scientific Q Exactive Plus. As an abbreviation for the ionized molecule [M]$^+$ or [M + H]$^+$ was used. Calibration was carried out using premixed calibration solutions (Thermo Fisher Scientific). The molecular fragments are stated as a ratio of mass per charge *m/z*.

## UV−Vis spectroscopy

UV−Vis spectra were recorded on a Lambda 750 (PerkinElmer) UV−Vis spectrophotometer, an UV/Vis/NIR spectrometer Cary 500 (Varian), and a Cary 3500 Multicell UV−Vis spectrophotometer. The absorbance was measured from 200 to 800 nm using a quartz cuvette of 2, 5, or 10 mm optical path length at 20 °C. According to the samples, the respective solvent was used as background and subtracted. If the sample contained metal ions, the metal ion was added to the blank in the same concentration as it was present in the sample. To dilute the samples, Eppendorf Research® plus pipets ranging from 2 to 20 µL, 20

to 200 µL, and 100 to 1000 µL and Hamilton glass syringes were used. The samples were prepared by weighing in the solid compounds and dissolving them in the appropriate solvents, followed by filtration using Fisherbrand PTFE syringe filters, 0.2 µm.

## High-performance liquid chromatography

Analytical high-performance liquid chromatography (HLPC) was performed using a Thermofisher UltiMate 3000 system containing a degasser, pump, autosampler, column compartment, and diode array detector. The flow rate was 1 mL/min on a stationary YMC-Triart C$_{18}$ ExRS plus column (8 nm, S−5 µm, 250 mm × 4.6 mm) or a VDSpher 100 C18-M-SE column (5 µm, 250 mm × 4.6 mm). Chromeleon 7 software was used for data extraction. Preparative HPLC was performed with the Puriflash® 4125 system from Interchim, equipped with InterSoft V5.1.08 software and a UV diode array detector (200−600 nm). The stationary phase was a preparative C$_{18}$ column (VDSpher 100 C18-M-SE, C18, 10 µm, 250 mm × 20 mm) at a flow rate of 15 mL/min. The precolumn was a small C$_{18}$ column (VDSpher 100 C18-M-SE, 10 µm, 20 × 10 mm).

## Procedure for irradiation

For irradiation experiments, the samples were dissolved in the respective solvent and placed in a translucent glass vial (clear glass crimp neck vial, borosilicate glass NMR tube, or Quartz cuvette). The vessel was then placed in a customized metal block, which has openings on the sides where the required LEDs are attached to irradiate the sample. Constant temperature of 22 ± 3 °C was maintained. Irradiation time, wavelengths, and power of the applied LEDs are listed in the supplementary information (Supplementary Table 1).

## Thermal stability

The thermal stability was determined for the respective photoisomers at 25 °C. After sample preparation, the solutions were irradiated with a wavelength that gives a high photoisomer content and then kept at 25 °C in the dark. The isomer ratio was determined by integration of the respective *(E)*- and *(Z)*-isomer signals in $^1$H NMR in intervals. The data was processed by calculating ln(X$_0$/X$_t$), where X is the percentage of the respective *(Z)*-isomer, followed by linear fitting (Eq. (1)). The calculated slope corresponds to the degradation rate constant *k* which is used to calculate the half-life $t_{1/2}$.

$$X_t = X_0 \times e^{-k \times t} \leftrightarrow \ln\left(\frac{X_0}{X_t}\right) = k \times t \tag{1}$$

## Photostationary states

The photostationary states were calculated according to Eq. (2), where $I_E$ is the integral of signals assigned to the *(E)*-isomer, and $I_Z$ the integral of respective signals of the *(Z)*-isomer.

$$\%((E)\ \text{isomer}) = \frac{I_E}{(I_E + I_Z)} \times 100 \tag{2}$$

## Photostability measurements

To evaluate photostability, the compound was irradiated at an appropriate wavelength to induce isomerization until the photostationary state was reached (evident by no further changes in the absorption spectrum), yielding a high proportion of the photoisomer. Subsequent irradiation at a shorter wavelength reversed the process, regenerating a high *(E)*-content. This photoswitching cycle was repeated ten times for each cage compound and the absorbance at λ$_{max,abs}$ was compared after the first and the last irradiation cycle. Experimental details for each compound are described in the respective

section in the manuscript. Irradiation intensities and irradiation time are summarized in Supplementary Table 1.

## LED irradiation

Sample irradiation for photoisomerization was performed using LEDs from Avonec, Mouser Electronics Inc., and OSRAM GmbH with emission maxima of 730, 700, 660, 623, 590, 523, 490, 470, 455, 430, and 365 nm. Standard time of irradiation for NMR and UV–Vis experiments are summarized in Supplementary Table 1. Any deviations from these standards are noted in the corresponding experiment. For the time of irradiation, samples were maintained at constant temperature ($22 \pm 3\,°C$) using a metal cooling block and a fan. Irradiation intensities of the respective LEDs were determined using the PowerMax USB (type PS19Q) sensor device (Coherent®) in five independent measurements. The detector (diameter 19 mm) was located at a distance of 55 mm from the light source, identical to the position of irradiated samples. The results are presented in Supplementary Table 1. "Max Power" refers to the smallest possible distance of the irradiated sample to the LED and "Min Power" to the largest possible distance between sample and LED.

## $CO_2$-mediated extraction of the cage 5 into water

The cage 5 is insoluble in water in its neutral form, but upon protonation, it becomes well-soluble in water and can be fully transferred into the aqueous phase. 5 was dissolved in chloroform to form 1 mM solution, and then mixed with $D_2O$, resulting in no extraction into the water phase. A constant stream of $CO_2$ bubbled through the solution for 10 min resulted in the transfer of protonated 5 (5*H⁺) into the water phase. Stirring that system for 30 h in an open vessel reversed the process (due to $CO_2$ evaporation into the atmosphere), bringing the deprotonated 5 back into the organic layer.

## Crystallographic study

Single crystals were obtained by a vapor-diffusion system of chloroform/n-hexane, whereby the compound was dissolved in chloroform at c = 0.5–2 mM in an open vial and then placed in a larger vessel containing n-hexane. The outer vessel was sealed and placed in the fridge at $3\,°C$. Single crystal X-ray diffraction data were collected on a STOE STADI VARI diffractometer with monochromated Ga Kα (1.34143 Å) radiation generated by an EXCILLIUM METALJET D2+ at low temperature. Using Olex2[35], the structures were solved with the ShelXT[36] structure solution program using Intrinsic Phasing and refined with the ShelXL[37] refinement package using Least Squares minimization. Refinement was performed with anisotropic temperature factors for all non-hydrogen atoms; hydrogen atoms were calculated on idealized positions.

## Crystal data

*(E,E,E)*-3: triclinic, space group $P\bar{1}$, a = 12.1445(5) Å, b = 14.1652(5) Å, c = 16.8733(6) Å, α = 71.636(3)°, β = 87.999(3)°, γ = 68.328(3)°, V = 2549.21(18) Å³, T = 180 K, Z = 2, $\rho_{calc}$ = 1.453 g/cm³, $R_1$ = 0.0620, $wR_2$ = 0.0902, CCDC number = 2448723.

*(E,E,E)*−5: monoclinic, space group $P2_1/c$, a = 21.2547(16) Å, b = 11.3016(5) Å, c = 23.2772(16) Å, α = γ = 90°, β = 100.134(6)°, V = 5504.2(6) Å³, T = 180 K, Z = 4, $\rho_{calc}$ = 1.412 g/cm³, $R_1$ = 0.1486, $wR_2$ = 0.2486, CCDC number = 2448724.

## MTT assays

HeLa cells were cultured in Dulbecco's Modified Eagle Medium (DMEM) supplemented with 10% fetal calf serum (FCS) and 1% penicillin/streptomycin solution (Gibco®, 10000 units/mL penicillin and 10000 μg/mL streptomycin) in a humid incubator at $37\,°C$ and 5% $CO_2$. To suspend the cells, they were first washed with PBS (Gibco®) and then treated with trypsin-EDTA (0.25%, Gibco®) for 5 min to detach them from the surface. Flat-bottom 96-well plates were prepared by

adding 200 μL/well PBS to the outer border wells to maintain constant vaporization in all wells. The cells were seeded into the inner wells (3000 cells/well) and incubated for 24 h. Stock solutions (500 μM) of compound 5*H⁺ were prepared in $H_2O$, and aliquots of the solutions were irradiated with 523 nm for 1 h (10 W LED from LED ENGIN; maximum power = 7.08 mW/cm², minimum power = 6.56 mW/cm²; see Supplementary Table 1). The samples were subsequently diluted with DMEM to the final concentrations (0.091 μM, 0.5 μM, 1 μM, 2 μM, 4 μM, 6 μM, 8 μM, and 10 μM, 2% $H_2O$, v/v). The medium was replaced with medium supplemented with 5*H⁺ (dark or irradiated) in different concentrations (100 μL/well, six technical replicates), and the cells were incubated for another 48 h in the dark. To ensure uniform treatment of the cells, the medium of the controls was replaced with DMEM supplemented with 2% $H_2O$ (v/v, 100 μL/well). After the 48-h incubation period, 5 μL/well Triton™ X-100 detergent (10% solution in PBS (w/v)) was added to the positive control for at least 5 min to induce cell death. Subsequently, 10 μL/well MTT dye solution in water (3-(4,5-dimethylthiazol-2-yl)-2,5-diphenyltetrazolium bromide, Cell Proliferation Kit I from Roche) was added to all sample wells and cells were incubated for 3 h. The reduction of the dye to formazan was stopped by adding 100 μL/well of solubilization buffer (Cell Proliferation Kit I from Roche), and the plates were placed in the incubator overnight to ensure solubilization of formazan. The absorbance of each well was measured at 595 nm using a CLARIOstar Plus plate reader (BMG Labtech). The absorbance data were averaged across the technical replicates, the positive control was subtracted as background, and the data were then normalized to the number of viable cells in the negative control as 100%. Data were plotted against the log of agonist concentration ($\log_{10}$[agonist], M) with mean and SD in GraphPad Prism Version 9.5.1 for Windows, GraphPad Software, San Diego, California, USA. The number of individual experiments n performed for the dark and irradiated samples was n = 3. The HeLa cell line was obtained from DSMZ (Leibniz-Institut Deutsche Sammlung von Mikroorganismen und Zellkulturen), DSMZ no. ACC 57; deposited by ATCC (CCL 2), Rockville, Maryland, USA.

**Positive control**. Cells that were specifically killed by treating them with 5 μL/well Triton™ X-100 detergent (10% solution in PBS (w/v)) for at least 5 min before adding MTT dye.

**Negative control**. Cells that remained alive (treated with neither 5*H⁺ nor Triton™ X-100), corresponding to 100% cell viability.

## Reporting summary

Further information on research design is available in the Nature Portfolio Reporting Summary linked to this article.

# Data availability

All the data supporting the findings of this study (such as the data relating to the "Methods", experimental procedures, NMR, MS, and UV–Vis spectra) are available within the paper and its Supplementary Information files, including the raw data from cell viability assays (MTT assays, Source Data). The X-ray crystallographic coordinates for structures reported in this study: the cages *(E,E,E)*-3 and *(E,E,E)*−5 as well as the compounds *(E)*-6, *(E)*-1 and *(E)*-9 have been deposited at the Cambridge Crystallographic Data Centre (CCDC), under deposition numbers 2448723 (*(E,E,E)*-3), 2448724 (*(E,E,E)*−5), 2448725 (*(E)*-6), 2448726 (*(E)*-1), 2448727(*(E)*-9). These data can be obtained free of charge from The Cambridge Crystallographic Data Centre (www.ccdc.cam.ac.uk) via www.ccdc.cam.ac.uk/data_request/cif. Source data are provided with this paper.

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

## Acknowledgements

The authors gratefully acknowledge the financial support from Deutsche Forschungsgemeinschaft (DFG)—the grants PI 1124/6-3, PI 1124/12-1, PI 1124/15-1, and GRK 2039/1 (Z.L.P.), YIN Grant of KIT Karlsruhe (Z.L.P.), the Land of Baden-Württemberg for the support in the form of Landesgraduiertenstipendium (V.S.), EPICUR SEED Funding SusPhotoPoly (A.S and Z.L.P.), and Evonik Stiftung for the Doctoral Fellowship (A.S.). The authors gratefully acknowledge the infrastructural support of our research by Prof. Dr. Stefan Bräse (KIT Karlsruhe). We want to thank Prof. Dr. Hans-Achim Wagenknecht for providing us access to his equipment. The authors acknowledge support by the state of Baden-Württemberg through bwHPC (bw19J002) and the German Research Foundation (DFG) through grant no. INST 40/575-1 FUGG (JUSTUS 2 cluster), as well as the KIT Publication Fund of the Karlsruhe Institute of Technology.

## Author contributions

V.S.: synthesis, photophysical characterization, methodology, data analysis, and writing. A.S.: cell viability assays (MTT assays). O.F.: crystal structure determination. Z.L.P.: conceptualization,

methodology, supervision, funding acquisition, writing, and project administration.

## Funding

## Competing interests
The authors declare no competing interests.
