## [Transparent Peer Review file · Nature Communications]

Dynamically Assembled Photochromic Cages Operational in Water with Visible Light

Corresponding Author: Dr Zbigniew Pianowski

Version 0:

Reviewer comments:

Reviewer #1

(Remarks to the Author)

In this manuscript, Schäfer, Fuhr and Pianowski report a series of photoswitchable cages that are able to operate in water. The authors make use of one of their previously reported photochromic motifs (a tetra-ortho-fluoroazobenzene bis-aldehyde) as well as a new motif being tetra-ortho-chloroazobenzene (aiming to further induce a red-shift in the E-isomer UV/vis spectra). The tetra-ortho-fluorinated switch was able to be assembled with TREN to form a [3+2] capsule. The authors could convert this EEE-capsule to the ZZZ almost quantitatively with 660 nm light with no mixed intermediate EEZ or EZZ states being observed (attributed to templating effect), but with a reduced thermal stability (although measured in CDCl₃ where trace amounts of acid could catalyse Z-to-E isomerism). The authors could reduce their dynamic-covalent imine bonds to secondary amines, affording more robust capsules which could also be photoisomerized, but using higher energy light due to the loss in conjugation. Protonation of this reduced cage renders the system water soluble, and the authors can display phase-transfer of the system between CDCl₃ and D₂O.

What is quite surprising is that the protonated form is also photoswitchable from the E to the Z-isomer. This enabled the authors to use a transient stimulus (carbonic acid) to develop a cycle that protonates the capsule (transferring it from CDCl₃ to D₂O), and then CO₂ is given off over time resetting the system back to the chlorinated phase. This could be particularly interesting if any Host-Guest chemistry could be achieved, allowing cargo transport between phases.

The authors further looked at the dynamic-covalent chemistry associated with the ZZZ stage of the imine capsule. They noted that exchange of a tetra-fluorinated bisaldehyde with benzaldehyde was possible. This is interesting as such transamination is typically observed for an amine reacting with the imine, but here it appears that the aldehyde component can also do this. The authors should clarify their proposed mechanism for this, or if they think there is some disassembly and reassembly taking place.

The thermal relaxation of ZZZ-3 to EEE-3 does not show any observable mixed ZZE or ZEE states, this is also quite an interesting result that sets it apart from previous work of Greenaway. It would be useful if the authors could further comment on this aspect of templation and why it is more significant for this tetra-ortho-fluorinated ligand. Further study into the use of bisaldehydes have been reported, possibly affording oligomeric structures.

The chlorinated photoswitch derivative shows reduced thermal half-life but ability to operate with near IR light. The resulting capsule appears unstable when converted to the ZZZ state, affording a mixture of species, but this could be regenerated by switching them back to the EEE state. This chlorinated cage could not be reduced with NaBH₄ (perhaps BH₃.THF would work?)

Finally, they investigate binding metal ions by using a bipyridine pillar. They can form complexes with various metal ions.

The experimental details are accurately reported, and the authors discuss the previous work in this area citing the most relevant papers.

Overall, I think this is a fantastic and full piece of work. I strongly congratulate the authors on their efforts, and I strongly support publication of this manuscript in Nature Communications, pending answers to some of the questions raised above and the minor points below.

Minor:

1. How does the thermal half-life of the cage 3 change when in DCM? Trace amounts of acid in chloroform has been known to catalyse Z-to-E isomerism.
2. For the NIR irradiation, the absorbance of the chlorinated switch above 700 nm looks minimal, how do the authors know that this is a switching effect and not a heating effect under such prolonged irradiation? Especially with such a short thermal

half-life.

Reviewer #2

(Remarks to the Author)

This manuscript describes the design, synthesis, and characterization of photoresponsive dynamic covalent cages assembled from azobenzene-derived bis-aldehydes and a trivalent amine linker. The authors demonstrate cages that undergo constitutional changes in response to light, pH, and metal ions. While the synthetic chemistry and characterization efforts are technically sound, the manuscript falls significantly short of the novelty, depth, and rigor expected for Nature Communications. I recommend rejection based on the following critical issues:

- A) The manuscript reads as a disjointed series of experiments rather than a well-packed study. It jumps between unrelated systems (fluorinated cage 3, reduced cage 5, chlorinated cage 10, bipyridine cage 14) without establishing a clear overarching hypothesis. The connection between individual results and the purported goal of "biocompatible molecular machines" is poorly articulated, weakening the conceptual impact.
- B) The core concept—light-driven dynamic covalent cages—and many specific methodologies (e.g., imine-based cage assembly, photoisomerization-triggered transformation, reduction to stabilize cages) have been extensively reported by others (e.g., Feringa [Ref 26], Clever [Ref 17], Schmidt [Ref 24], Wang [Ref 31]). The new chlorinated switch 9 is promising but underdeveloped; its integration into functional cages (10) is inefficient and lacks demonstration of significant advantages over prior systems. Therefore, the overall novelty of this work does not meet the general standard of Nature Communications.
- C) The title prominently claims cages "operational in water" implying broad applicability. However, detailed water-compatible studies are only demonstrated for the single, non-dynamic cage 5 (derived from 3 via reduction). The primary dynamic cages (3, 10, heteroleptic cages) are synthesized, switched, and characterized exclusively in organic solvents. Switching in water is only shown for cage 5 (after protonation). Even here, the "biocompatibility" claim is unsupported (no stability data in physiologically relevant media, no biological testing). The CO₂-triggered phase transfer of 5 is a simple pH effect, not a demonstration of "operation" as a functional machine in water. The title and abstract significantly overstate the actual aqueous results presented. In addition, protonation of azobenzene can induce Z to E transition; however, there is no comment for this effect (see ref J. Am. Chem. Soc. 2024, 146, 10246–10250).
- D) Key conclusions rely heavily on qualitative data (e.g., color changes, simple extractions, basic NMR spectra) and lack rigorous quantitative support. Claims of efficient self-assembly, exchange, and reversibility lack detailed kinetic data, thermodynamic parameters (e.g., binding constants for guest/metal interaction), or robust quantification of species distribution during stimuli-responsive processes (beyond limited NMR integrations). While X-ray structures of 3 and 5 are valuable, solution-state structural integrity (e.g., under switching conditions, in water) is not thoroughly established (e.g., lack of detailed DOSY for key species in relevant solvents/conditions).

Reviewer #3

(Remarks to the Author)

The manuscript of Pianowski et al. focuses on the synthesis and investigation of (non-)photoresponsive supramolecular cage made of substituted azobenzenes, imine/amine and bipyridine units. The achieved photoswitching within the therapeutic window of light along with the performed structural modifications towards water soluble cage are interesting. After reading the manuscript, we have raised the following questions that should be addressed prior to publication.

Page 1, the term "more recently developed structures" sounds awkward and should be replaced with concrete derivatives. Figure 1A, the structures of 1-4 shown should be completed with the elements to increase the chemical clarity. The NMR spectra should be completed with the integral intensities to properly distinguish the ration of 1 and 4

Page 4, Besides power of the used LEDs (given in the ESI), the authors should also specify the time of irradiation.

A comparison of the achieved fundamental parameters (e.g., half life of the Z isomeric form, used operational light wavelengths for photoswitching or maximal reached E/Z ratio) with known and structurally-related cages is missing.

Page 4, The text states that (Z,Z,Z) 5 returns to (E,E,E) 5 upon irradiation with 430 or 410 nm light. But the "430 nm" and "410 nm" NMR spectra in Figure S7 clearly contain more signals than the dark adapted photostationary states.

Page 4, second paragraph, Some references are missing or matched incorrectly, please check.

Why are not the photoswitching processes of 5 demonstrated by UV Vis spectroscopy as shown for 3 (Figure S43) and 10 (Figure S44)?

Figure 3, The double-arrow is used incorrectly.

Page 7, Replace "frequencies" with wavelengths, especially when discussing in nm.

Figure S34, The obtained value of 441.22 Da below the illustration of cage 14 does not match the theoretical value.

Page 11, The complexation of 14 with metals is referenced to Figure 6B but 3/13 are mentioned in the caption of Figure 6B. Extended data Figure 1A, A continuous irradiation with 660 nm light is mentioned, which contradicts Figure S5 mentioning "short intervals", what is correct?

Extended data Figure 2, How is the reaction of (Z,Z,Z) 3 with monovalent benzaldehyde or benzylamine affected by irradiation? Would the same result be delivered upon pre-irradiating cage 3 and subsequent reaction with monovalent reagents?

The fatigue resistance of the investigated photoresponsive cages should be completed, e.g. after 10 cycles.

Nothing is disclosed on how the trans. metal cations are bonded to the bipyridine moieties of 14?

The text should uniformly use stereochemical descriptors, e.g. (E,E,E)-5 as compared to often used E,E,E-5 (round brackets missing).

Reviewer #4

(Remarks to the Author)

Version 1:

Reviewer comments:

Reviewer #1

(Remarks to the Author)

The authors have carefully and fully considered my comments. I appreciate their willingness to constructively engage and I am pleased with the revisions made. I have no outstanding comments and I congratulate the authors on a nice and thorough study. I fully support publication of this manuscript in Nature Communications.

Reviewer #3

(Remarks to the Author)

The authors have addressed our comments and, therefore, we recommend publication.

Reviewer #4

(Remarks to the Author)

Reviewer #5

(Remarks to the Author)

In this manuscript, Pianowski and coworkers report a series of photoswitchable cages assembled from the bisaldehydes substituted azobenzene and a trivalent amine linker. It describes the design, synthesis, characterization of these photoresponsive cages and their response to the stimuli of light, pH, and metal ions.

From my point of view, great improvement has been made in the revised manuscript:

- 1) most of the issues raised by the three previous reviewers have been carefully addressed.
- 2) some additional experiments have been performed, including the synthesis of a novel stable covalent cage "13" by replacing the corresponding fluorinated bis-aldehyde component with their corresponding chlorinated one. This new cage can undergo a reversible photoisomerization in water under a 660 nm light irradiation condition with good biocompatibilities. Therefore, I would like to support the publication of this manuscript in Nature Communications after further addressing some of the following issues.
 - 1) When addressing the 4th question raised by reviewer 1 (regarding the high selective thermal relaxation of ZZZ-3 to EEE-3 without any observable mixed ZZE or ZEE states), the authors attributed it to the much shorter lifetimes of (E,E,Z)- and (E,Z,Z)-3 in comparison with the corresponding (E,E,E)- and (Z,Z,Z)-3 isomers. Since the formation of minor amounts of other isomers have been detected, I would suggest the authors to perform additional theoretical calculations (energy of the intermediates and/or final products and the favorite conformations, etc.) for a thorough explanation of the selectivity and stability difference of these isomers.
 - 2) When addressing the 8th question raised by reviewer 1 (regarding the heating effect on the switching of the isomers under a prolonged NIR irradiation condition, especially when considering that the absorbance of the chlorinated switch above 700 nm looks minimal), the authors used an aluminum-foil-wrapped sample as a control and irradiated under the same conditions as their main sample. In the tube covered with aluminum foil, they did not observe any isomerization under the same irradiation parameters (time, frequency of the used LED light emitter. It is highly suspicious to use an aluminum-foil-wrapped sample as a control, since this aluminum foil might reflect the NIR light, resulting in the lack enough energy to heat the sample. Some additional experiments should be provided to eliminate the thermal effect on the switching.
 - 3) When addressing Question D raised by reviewer 2 and the 26th question raised by reviewer 3 (regarding the binding constants for guest/metal interaction), the authors only provided some related literature and declared that they were not able to determine the binding constants for the cage "15" due to the limited amount of the compound. I would recommend the absorption titration to ascertain the binding mechanism of metal ions complexes with guest molecules, since this absorption titration only needs small amount of sample to get the binding constants.

Version 2:

Reviewer comments:

Reviewer #5

(Remarks to the Author)

The authors have carefully addressed my comments. Therefore, I would like to support the publication of this manuscript in Nature Communications.

REVIEWER COMMENTS

In the name of all authors, we would like to thank all the reviewers for their time and attention in reading our manuscript. We are grateful for all the comments, also the critical ones, which motivated us to increase the quality of our initial submission. The comments are individually addressed below.

Reviewer #1 (Remarks to the Author):

In this manuscript, Schäfer, Fuhr and Pianowski report a series of photoswitchable cages that are able to operate in water. The authors make use of one of their previously reported photochromic motifs (a tetra-ortho-fluoroazobenzene bis-aldehyde) as well as a new motif being tetra-ortho-chloroazobenzene (aiming to further induce a red-shift in the E-isomer UV/vis spectra). The tetra-ortho-fluorinated switch was able to be assembled with TREN to form a [3+2] capsule. The authors could convert this EEE-capsule to the ZZZ almost quantitatively with 660 nm light with no mixed intermediate EEZ or EZZ states being observed (attributed to templating effect), but with a reduced thermal stability (although measured in CDCl₃ where trace amounts of acid could catalyse Z-to-E isomerism). The authors could reduce their dynamic-covalent imine bonds to secondary amines, affording more robust capsules which could also be photoisomerized, but using higher energy light due to the loss in conjugation. Protonation of this reduced cage renders the system water soluble, and the authors can display phase-transfer of the system between CDCl₃ and D₂O.

1) What is quite surprising is that the protonated form is also photoswitchable from the E to the Z-isomer.

Ad 1) Indeed. However, we have confirmed it by ¹H and ¹⁹F NMR and multiple switching cycles (NMR, UV-Vis) (**Table S2, Figure S12-S14, Figure S72, Figure S77-S78**).

2) This enabled the authors to use a transient stimulus (carbonic acid) to develop a cycle that protonates the capsule (transferring it from CDCl₃ to D₂O), and then CO₂ is given off over time resetting the system back to the chlorinated phase. This could be particularly interesting if any Host-Guest chemistry could be achieved, allowing cargo transport between phases.

Ad 2) We have tried to encapsulate over 20 potential guest molecules (such as phenols, naphthols, aromatic sulphonic acids and halogenated aromatic compounds). However, we could not see any convincing proof of encapsulation, nor of its photomodulation, probably due to small cavity size. On the other hand, as the shape of the cage changes upon photoisomerization, the protonated cage **5** can interact as a guest with cucurbit[8]uril (CB8).

In the resulting inclusion complex we could see an influence on the isomer distribution in comparison to the cage dissolved in water in the absence of CB8, especially after irradiation at 410 nm (**Figures S20-S25** and **Table S3**).

3) The authors further looked at the dynamic-covalent chemistry associated with the ZZZ stage of the imine capsule. They noted that exchange of a tetra-fluorinated bisaldehyde with benzaldehyde was possible. This is interesting as such transimination is typically observed for an amine reacting with the imine, but here it appears that the aldehyde component can also do this. The authors should clarify their proposed mechanism for this, or if they think there is some disassembly and reassembly taking place.

Ad 3) As depicted in **Figure 3**, we believe that due to the presence of residual humidity in the system (water was not actively eliminated from the reaction mixtures), the more strained (*Z,Z,Z*)-**3** isomer partially opens to the form that contains a free aldehyde and an amine groups, which both can then interact with externally added reactive partners and result in the aldehyde / imine exchange. Additionally, the experiments were performed in CDCl₃, so the minor acidity of chloroform may also help catalyze this process.

4) The thermal relaxation of ZZZ-**3** to EEE-**3** does not show any observable mixed ZZE or ZEE states, this is also quite an interesting result that sets it apart from previous work of Greenaway. It would be useful if the authors could further comment on this aspect of templation and why it is more significant for this tetra-ortho-fluorinated ligand.

Ad 4) As this result is indeed interesting, the behavior of **3** upon irradiation with various light wavelengths has been thoroughly investigated. The interesting result, shown in the Figure 1B, has been confirmed in several independent experiments, to minimize the risk of error. Of course, the formation of other isomers is not impossible, but under the conditions applied by us, the fully converted isomers are evidently dominating. While 660 nm results in near-quantitative (*Z,Z,Z*)-**3** formation and 430 nm restores the (*E,E,E*)-isomer, 523 nm forms small quantities of the (*E,E,Z*)- and (*E,Z,Z*)-isomer, while the (*Z,Z,Z*)-isomer remains the main photoproduct (**Figure S2**). Yet, the intermediary forms have much shorter lifetimes (probably caused by the enhanced strain within the cage) than the (*Z,Z,Z*)-isomer (123 h at 25 °C, **Figure S3**). While after 523 nm irradiation, signals assigned to (*E,E,Z*)- and (*E,Z,Z*)-**3** are visible immediately after irradiation, they disappear within 1 day at 25 °C (**Figure S5**). While unequivocally assigning the signals to either (*E,E,Z*) or (*E,Z,Z*) was not possible, we have determined their lifetimes to be $t_{1/2} = 2.16$ h and $t_{1/2} = 6.54$ h (**Figure S6**).

Another recently published (H. Shan *et al. J. Am. Chem. Soc.* **2025**, *147*, 14960 – 14965, <https://doi.org/10.1021/jacs.5c04399>) cage system, based on azobenzenes, showed different switching behavior (either concerted *ZZZ*-to-*EEE* transition, or stepwise *ZZZ-EZZ-EEZ-EEE* switching) depending on the anion that is complexed in the adjacent ligand. Therefore, we find such a behavior rare, but not unprecedented.

In our case, we believe that the presence of fluorine atoms might additionally stabilize the (*E,E,E*)-**3** and (*Z,Z,Z*)-**3** isomers due to the favorable overlap of the C-F bonds or the fluorinated aryl residues, and thus sequester the intermediary forms from the system. As stated below, the analogous chlorinated cage (where the C-Cl bonds are less polarized) does not show such effect, forming various species and transient isomers upon irradiation (although the geometry in the latter case is also more strained due to the presence of bulk chlorine atoms, so this effects cannot be seen separately).

5) Further study into the use of bisaldehydes have been reported, possibly affording oligomeric structures.

The chlorinated photoswitch derivative shows reduced thermal half-life but ability to operate with near IR light. The resulting capsule appears unstable when converted to the *ZZZ* state, affording a mixture of species, but this could be regenerated by switching them back to the *EEE* state. This chlorinated cage could not be reduced with NaBH₄ (perhaps BH₃.THF would work?)

Ad 5) We thank the reviewer for motivating us to explore the issue of chlorinated cages even further. Although the BH₃*THF did not work in our system either (various products observed, but not the desired one), upon screening on alternative reaction conditions we discovered that other reducing agent - NaBH(OAc)₃ – can provide us with the desired product (the reduced, covalently bound and stable chlorinated cage **13**), although in moderate yield (we have isolated the expected product with 15 % overall yield, starting from the assembly of the dynamic cage **10** out of the component – the tetra-chloro-azobenzene bisaldehyde **9** and TREN **2**, and applying the reducing agent *in situ*).(**Figures S56-S57**). This cage **13** was switchable with red light (660 nm) and characterized further (**Figure S58-S60**; see below).

6) Finally, they investigate binding metal ions by using a bipyridine pillar. They can form complexes with various metal ions.

The experimental details are accurately reported, and the authors discuss the previous work in this area citing the most relevant papers. Overall, I think this is a fantastic and full piece of work. I strongly congratulate the authors on their efforts, and I strongly support publication of this manuscript in Nature Communications, pending answers to some of the questions raised above and the minor points below.

Ad 6) we highly appreciate the opinion of the reviewer on the novelty and interesting aspects of our manuscript

Minor:

7) How does the thermal half-life of the cage **3** change when in DCM? Trace amounts of acid in chloroform has been known to catalyse Z-to-E isomerism.

Ad 7) Indeed we observed a difference in thermal stability of the cage *(Z,Z,Z)*-**3**, with $t_{1/2}$ in dichloromethane being 321 h (**Figure S32**), compared to 123 h in CDCl_3 (**Figure S3**). In the non-acidic solvent, the life-time of the cage *(Z,Z,Z)*-**3** is now similar to the linear imine *(Z)*-**17** (307 h, **Figure S33**), but we still do not see the formation of intermediary isomers EEZ / EZZ, which supports our remarks above on the templating effect.

8) For the NIR irradiation, the absorbance of the chlorinated switch above 700 nm looks minimal, how do the authors know that this is a switching effect and not a heating effect under such prolonged irradiation? Especially with such a short thermal half-life.

Ad 8) especially with such minimal effects, we tried to perform careful control of the system. Primarily, for the 730 nm irradiation we have applied a 715-nm-cutoff filter, to make sure that we do not irradiate our sample with the emission peak shoulder in the red light range below 715 nm. To exclude the photothermal effect, which is known and often reported, particularly for the NIR-irradiated systems, we wrapped a control sample with aluminum foil and irradiated under the same conditions as our main sample. In the tube covered with aluminium foil, we did not observe any isomerization under the same irradiation parameters (time, frequency of the used LED light emitter. Thus, we attribute the observed isomerization purely to the photon absorption by the sample (**Figure S44**, including a photo of our experimental setup).

Reviewer #2 (Remarks to the Author):

This manuscript describes the design, synthesis, and characterization of photoresponsive dynamic covalent cages assembled from azobenzene-derived bis-aldehydes and a trivalent amine linker. The authors demonstrate cages that undergo constitutional changes in response

to light, pH, and metal ions. While the synthetic chemistry and characterization efforts are technically sound, the manuscript falls significantly short of the novelty, depth, and rigor expected for Nature Communications.

We appreciate the careful reading of our manuscript by the Reviewer 2 and his critical opinion on our work. It became a motivation for us to increase the quality of our manuscript, which we will discuss in details in our comments on particular critical issues below. In particular, we have provided additional experiments to demonstrate the range of biocompatibility of our compounds. Additionally, the upgraded version of our manuscript contains a novel stable covalent cage “13” based on chlorinated bis-aldehyde components, which undergoes reversible photoisomerization in water under irradiation with 660 nm light – the wavelength that efficiently penetrates (to the depth of several cm, according to the available literature) soft human tissues.

I recommend rejection based on the following critical issues:

A) The manuscript reads as a disjointed series of experiments rather than a well-packed study. It jumps between unrelated systems (fluorinated cage 3, reduced cage 5, chlorinated cage 10, bipyridine cage 14) without establishing a clear overarching hypothesis. The connection between individual results and the purported goal of "biocompatible molecular machines" is poorly articulated, weakening the conceptual impact.

Ad A) The clear overarching hypothesis in the manuscript is that we can synthesize - with good yields and good control over their constitution - complex photochromic molecular cages using simple methodology demonstrated in the manuscript. The resulting dynamic cages can undergo the exchange with numerous components in a light-controlled fashion. Yet, upon reduction we obtain stable compounds (in the revised version, two cages “5” and “13”) capable of photoisomerization in aqueous conditions with biocompatible light frequencies (see below). All the presented dynamic systems have similar topology and therefore cannot be treated as disjointed. The latter series of experiments build up on the initial observations regarding the synthesis and behavior of the dynamic cage “3” and are directly related/comparable with its design and properties. The latter series intent to demonstrate the vast flexibility and stimuli-responsiveness of the initial setup. The additional highlight is the power of supramolecular

self-assembly, as the stepwise synthesis of cage “5” and its chlorinated analogue “13” would be extremely challenging.

To make the connection of our results to the overarching concept more clear, and at the same time to underline that our system does not constitute yet a biocompatible molecular machine, but is an important step towards realizing that goal, the following sentence in conclusions has been given as follows: “We believe, that especially our cages operational in aqueous media with red light make a step forward towards construction of biocompatible molecular machines”

B) The core concept—light-driven dynamic covalent cages—and many specific methodologies (e.g., imine-based cage assembly, photoisomerization-triggered transformation, reduction to stabilize cages) have been extensively reported by others (e.g., Feringa [Ref 26], Clever [Ref 17], Schmidt [Ref 24], Wang [Ref 31]). The new chlorinated switch 9 is promising but underdeveloped; its integration into functional cages (10) is inefficient and lacks demonstration of significant advantages over prior systems. Therefore, the overall novelty of this work does not meet the general standard of Nature Communications.

Ad B) We tried our best to present the state-of-the-art and credit our predecessors properly (as listed by the Reviewer 2 above in “reported by others”). We sincerely hope that we did not miss any important contribution to the area. However, in our manuscript a novel quality emerged out of existing methodologies in several aspects listed below.

In the first version, we reported an unprecedented observation of clean photoconversion between two oppositely configured photoisomers: *(Z,Z,Z)*-3 and the *(E,E,E)*-3, while our forerunners typically report pretty complex mixtures of isomers (geometry and constitutional) at photoequilibria, often including significant fractions of linear polymers which irreversibly precipitate out of the solution. While we were certainly lucky with the choice of substrates and the reaction conditions (as this was one of our first experiments, and not the result of lengthy optimization), this does not diminish the novelty of our observation and the progress it makes to the methodology.

We agree with the Reviewer 2, that the cage formation from the chlorinated switch “9” was promising, but underdeveloped in the first version of the manuscript. We are happy to report that in the revised version we managed to provide a covalent cage “13” containing the chlorinated switch 9. This cage is reversibly switchable in water with red light (660 nm) (see **Figures S58 and S60**) – a frequency that can deeply penetrate soft human tissues. A cage reversibly switchable in water with red light was – to our best knowledge - not demonstrated by others, and we believe that this increases the novelty of our report even further, particularly

in its aspect of biocompatibility. We believe that it shows the way (to us and others) to design more complex systems with enhanced functionalities – and, in perspective, true molecular machines - that will be operational *in vivo*.

C) The title prominently claims cages "operational in water" implying broad applicability. However, detailed water-compatible studies are only demonstrated for the single, non-dynamic cage **5** (derived from **3** via reduction). The primary dynamic cages (**3**, **10**, heteroleptic cages) are synthesized, switched, and characterized exclusively in organic solvents. Switching in water is only shown for cage **5** (after protonation). Even here, the "biocompatibility" claim is unsupported (no stability data in physiologically relevant media, no biological testing). The CO₂-triggered phase transfer of **5** is a simple pH effect, not a demonstration of "operation" as a functional machine in water.

Ad C) In the revised manuscript, we provided a second non-dynamic cage "**13**" – the chlorinated analogue of **5** – which is also switchable in water, now with 660 nm red light. Of course, as the reviewer correctly remarked, the term "biocompatibility" was in our case related mainly to the used wavelength of light, and – in the broader sense – it encompasses multiple factors. For that reason, we have tested the stability of the protonated cage "**5**" in physiologically relevant media - in water and in acetate buffer pH 3.6 (**Figures S77** and **S78**, respectively). After 10 switching cycles, we have seen less than 1% degradation, which attests to very good stability of "**5**" in aqueous media.

We have also tested the cytotoxicity of the cages **5** towards HeLa cells using a standard technique (MTT assays) (**Figure S26**, **Tables S4**). It demonstrated, that a) the cage **5** can be used in biological context at the concentration below 0.5 μ M, and b) that photoisomerization does not significantly alter the cytotoxicity of this cage. Due to the insufficient amount of material, we could not perform the MTT assays for the cage "**13**", but we have no reason to suppose that the toxicity would drastically differ from the fluorinated derivative "**5**", as the cage "**13**" does not have additional H-bonding sites, charged or ionizable groups such as amines/carboxylates, nor pharmacophore-related motifs.

Regarding the last issue ("operation of the machine in water"), while the cages do not form host-guest complexes with small cargo molecules (as far as the over 20 compounds screened by us indicated), we have observed that upon formation an inclusion complex of the cage **5** in water with cucurbit(8)uril, the photoisomerization output (upon irradiation with violet light) differs from the outcome in free solution. These results were added in the revised version (**Figures S18-S25**) and expand the scope of functionality of the demonstrated system.

C1) The title and abstract significantly overstate the actual aqueous results presented. In addition, protonation of azobenzene can induce Z to E transition; however, there is no comment for this effect (see ref J. Am. Chem. Soc. 2024, 146, 10246–10250).

Ad C1) We have analyzed the stability of the cage “**5**” in chloroform (in the neutral form) and in water (as the protonated system) (**Figures S15-S16**). In both cases, we followed by NMR the samples over the course of several weeks. We have not noticed any drastic impact of protonation on the lifetime of the (*Z,Z,Z*)-**5** cage.

D) Key conclusions rely heavily on qualitative data (e.g., color changes, simple extractions, basic NMR spectra) and lack rigorous quantitative support. Claims of efficient self-assembly, exchange, and reversibility lack detailed kinetic data, thermodynamic parameters (e.g., binding constants for guest/metal interaction), or robust quantification of species distribution during stimuli-responsive processes (beyond limited NMR integrations). While X-ray structures of **3** and **5** are valuable, solution-state structural integrity (e.g., under switching conditions, in water) is not thoroughly established (e.g., lack of detailed DOSY for key species in relevant solvents/conditions).

Ad D) we are happy to report that additional data (DOSY and other relevant 2D-NMR experiments, additional quantification of species’ distribution during stimuli-responsive processes) provided in the revised manuscript fully support our previous conclusions. We have provided detailed insight into the photoisomerization results under various conditions, and the time course of certain processes (instead of the start and end point provided in the initial version of the manuscript).

To be more detailed: we provided COSY, NOESY, and DOSY, spectra of the dark-state cage molecules and their photoisomers under various conditions (**Figures S80-S111**), including short discussion on hydrodynamic radii of the particular cages, which are in agreement with the observable properties in other analytical methods. In particular, the **Figures S110-S111** clearly supports our earlier discussion on partial disassembly of the dynamic chlorinated cage “**10**” upon irradiation with 660 nm. We quantified photostability of the cages under various conditions over 20 irradiation steps (**Figure S74-S79**), specified the photostationary states of **5***H⁺ in water (**Table S2**), and determined *k* and *t*_{1/2} for the other isomers of **3** (*(E,E,Z)* and *(E,Z,Z)*) and for (*Z,Z,Z*)-**3** and (*Z*)-**17** in DCM. A summary of fundamental parameters is given in **Extended Data Table 1**.

Due to the low amount of the material, we were not able to determine the metal ion binding constants for the cage “**15**”. However, as such non-photochromic complexes (and similar ones) have been previously reported in literature, we have added related references (refs. [32-34]) for the readers who are interested in such architecture in more details.

Reviewer #3 (Remarks to the Author):

The manuscript of Pianowski et al. focuses on the synthesis and investigation of (non-)photoresponsive supramolecular cage made of substituted azobenzenes, imine/amine and bipyridine units. The achieved photoswitching within the therapeutic window of light along with the performed structural modifications towards water soluble cage are interesting.

We thank the Reviewer 3 for his time, questions and suggestions which help us to improve the quality of our manuscript.

After reading the manuscript, we have raised the following questions that should be addressed prior to publication.

11) Page 1, the term “more recently developed structures“ sounds awkward and should be replaced with concrete derivatives.

Ad 11) Concrete derivatives have been placed in the manuscript

12) Figure 1A, the structures of 1-4 shown should be completed with the elements to increase the chemical clarity.

Ad 12) We have added the chemical structures to Figure 1A and Figure 2 for more clarity.

13) The NMR spectra should be completed with the integral intensities to properly distinguish the ration of 1 and 4

Ad 13) The integrations in Figure 1 A have been added

14) Page 4, Besides power of the used LEDs (given in the ESI), the authors should also specify the time of irradiation.

Ad 14) We have specified irradiation times explicitly for each figure in the main part. In supporting information, standardized protocols include the irradiation times have been added (**Table S1**). Any deviations have been marked in the respective experiment descriptions. Otherwise they follow the general protocols.

15) A comparison of the achieved fundamental parameters (e.g., half life of the Z isomeric form, used operational light wavelengths for photoswitching or maximal reached E/Z ratio) with known and structurally-related cages is missing.

Ad 15) We have added parameters for several structurally related cages from literature (half-life times, absorbance maxima and photoconversion) and summarized our results for comparison in **Extended Data Table 1**);

16) Page 4, The text states that (Z,Z,Z) **5** returns to (E,E,E) **5** upon irradiation with 430 or 410 nm light. But the “430 nm” and “410 nm” NMR spectra in Figure S7 clearly contain more signals than the dark adapted photostationary states.

Ad 16) We thank to the reviewer for this remark. Indeed, the conversion is not quantitative for the reduced cage **5**. We have rephrased the respective sentences to accurately reflect the isomer distribution, listed in the **Table S2** and on the **Figure S9-S10 and Figure S12-S13**) (in chloroform and – protonated – in water).

17) Page 4, second paragraph, Some references are missing or matched incorrectly, please check.

Ad 17) we will appreciate, if the reviewer can tell us more specifically which references need to be changed/added, as the second paragraph of the page 4 (the section entitled “Reduction of **3** yields a stable cage **5** reversibly switchable in water”) contains no references at all. Maybe the reviewer had in mind another section. If so, we will be happy to correct it. After preventive checking, we have found no incorrectly matched references among the existing ones.

18) Why are not the photoswitching processes of **5** demonstrated by UV Vis spectroscopy as shown for **3** (Figure S43) and **10** (Figure S44)?

Ad 18) We have provided the UV-Vis spectra illustrating the photoswitching process of **5** and **5*H⁺** (**Figures S71-S72**).

19) Figure 3, The double-arrow is used incorrectly.

Ad 19) Indeed, by mistake we have used the wrong arrow- the arrow has been corrected and the required water molecule has been added.

20) Page 7, Replace “frequencies” with wavelengths, especially when discussing in nm.

Ad 20) The word frequencies was exchanged to wavelengths whenever in context with nm.

21) Figure S34, The obtained value of 441.22 Da below the illustration of cage **14** does not match the theoretical value.

Ad 21) We have checked the Figure S34 (now **Figure S61**) and it seems to contain the correct value 411.22 Da.

22) Page 11, The complexation of 14 with metals is referenced to Figure 6B but 3/13 are mentioned in the caption of Figure 6B.

Ad 22) Due to the addition of an extra cage (the chlorinated covalent cage, now with the number “**13**”), the numbers above have been shifted by one. In the old version, the cage **14** is formed insitu upon mixing of the pillar **13** with TREN or with the cage **3**. Therefore, the components of the experiment (**3 / 13**) are mentioned in the caption. We have double-checked the content of the caption, and it corresponds to the composition of respective experiments, to the best of our knowledge. Now the bipyridine-cage will have the number **15**, and the respective pillar – number **14**. So in the caption it will be cage **15** and the components **3/14**.

23) Extended data Figure 1A, A continuous irradiation with 660 nm light is mentioned, which contradicts Figure S5 mentioning “short intervals”, what is correct?

Ad 23) The correct description is “short intervals”. We have adapted the **Extended data Figure 1B** accordingly.

24) Extended data Figure 2, How is the reaction of (Z,Z,Z) **3** with monovalent benzaldehyde or benzylamine affected by irradiation? Would the same result be delivered upon pre-irradiating cage **3** and subsequent reaction with monovalent reagents?

Ad 24) We conducted the experiment with pre-irradiated cage **3** which resulted in a decreased conversion, especially for the reaction with benzaldehyde. This may be reasoned with the increasing formation of unreactive (Z,Z,Z)-**3** over the course of the experiment. Details have been added **Figure S27-S28** and **Table S5**.

25) The fatigue resistance of the investigated photoresponsive cages should be completed, e.g. after 10 cycles.

Ad 25) The fatigue resistances have been measured and provided in **Figure S74-S79**.

26) Nothing is disclosed on how the trans. metal cations are bonded to the bipyridine moieties of **14**?

Ad 26) The binding of metals cations to **14** (now **15**) have been clarified in the literature reference added by us to the revised version (ref.[32])

27) The text should uniformly use stereochemical descriptors, e.g. (E,E,E)-**5** as compared to often used E,E,E-**5** (round brackets missing)

Ad 27) We thank for this remark. Stereochemical descriptors have been unified across the paper.

REVIEWER COMMENTS

In the name of all authors, we would like to thank all the reviewers for their time and attention in reading our manuscript. We are grateful for all the comments, also the critical ones, which motivated us to increase the quality of our initial submission. The comments from reviewers 1-4 have been addressed in the previous revision round. The comments of the reviewer 5 are individually addressed below.

Reviewer #1 (Remarks to the Author):

The authors have carefully and fully considered my comments. I appreciate their willingness to constructively engage and I am pleased with the revisions made. I have no outstanding comments and I congratulate the authors on a nice and thorough study. I fully support publication of this manuscript in Nature Communications.

Reviewer #3 (Remarks to the Author):

The authors have addressed our comments and, therefore, we recommend publication.

Reviewer #4 (Remarks to the Author):

Reviewer #5 (Remarks to the Author):

In this manuscript, Pianowski and coworkers report a series of photoswitchable cages assembled from the bisaldehydes substituted azobenzene and a trivalent amine linker. It describes the design, synthesis, characterization of these photoresponsive cages and their response to the stimuli of light, pH, and metal ions. From my point of view, great improvement has been made in the revised manuscript: 1) most of the issues raised by the three previous reviewers have been carefully addressed. 2) some additional experiments have been performed, including the synthesis of a novel stable covalent cage "13" by replacing the corresponding fluorinated bis-aldehyde component with their corresponding chlorinated one. This new cage can undergo a reversible photoisomerization in water under a 660 nm light irradiation condition with good biocompatibilities. Therefore, I would like to support the publication of this manuscript in Nature Communications after further addressing some of the following issues.

1) When addressing the 4th question raised by reviewer 1 (regarding the high selective thermal relaxation of ZZZ-3 to EEE-3 without any observable mixed ZZE or ZEE states), the authors attributed it to the much shorter lifetimes of (E,E,Z)- and (E,Z,Z)-3 in comparison with the corresponding (E,E,E)- and (Z,Z,Z)-3 isomers. Since the formation of minor amounts of other isomers have been detected, I would suggest the authors to perform additional theoretical calculations (energy of the intermediates and/or final products and the favorite conformations, etc.) for a thorough explanation of the selectivity and stability difference of these isomers.

Ad 1) we appreciate the reviewer's insight into this issue. However, as we are a synthetic organic group with some capacities in molecular biology, we cannot guarantee that our theoretical calculations are performed correctly. The proper performance of that task would require additional collaboration or an effort that exceeds the scope of the current manuscript. The Editor supported this point of view and notified us that "*additional theoretical calculations are not required at this time.*"

2) When addressing the 8th question raised by reviewer 1 (regarding the heating effect on the switching of the isomers under a prolonged NIR irradiation condition, especially when considering that the absorbance of the chlorinated switch above 700 nm looks minimal), the authors used an aluminum-foil-wrapped sample as a control and irradiated under the same conditions as their main sample. In the tube covered with aluminum foil, they did not observe any isomerization under the same irradiation parameters (time, frequency of the used LED light emitter). It is highly suspicious to use an aluminum-foil-wrapped sample as a control, since this aluminum foil might reflect the NIR light, resulting in the lack enough energy to heat the sample. Some additional experiments should be provided to eliminate the thermal effect on the switching.

Ad 2) We thank the reviewer for bringing potential NIR-reflection problems with our aluminum foil control sample to our attention. Thus we revised the original setup and used a black vial (absorbing visible and NIR light) as a control sample instead of one wrapped in aluminum foil during 730 nm irradiation (**Figure S44A**). We also monitored the temperature of a sample of **9** during 730 nm irradiation and we monitored the isomeric ratio of **9** at 30 °C, 40 °C and 50 °C (**Figure S44B**). Since neither the sample in the black vial, nor the samples at elevated temperatures formed any (*Z*)-isomer, we concluded that indeed, 730 nm absorption is responsible for the observed (*E*)→(*Z*)-isomerization (see **Figure S44** and respective description for a more detailed discussion).

3) When addressing Question D raised by reviewer 2 and the 26th question raised by reviewer 3 (regarding the binding constants for guest/metal interaction), the authors only provided some related literature and declared that they were not able to determine the binding constants for the cage "15" due to the limited amount of the compound. I would recommend the absorption titration to ascertain the binding mechanism of metal ions complexes with guest molecules, since this absorption titration only needs small amount of sample to get the binding constants.

Ad 3) We would like to thank the reviewer for encouraging us to examine cage **15** and its interaction with Fe²⁺ in more detail. We were able to isolate **15** in good yield (see added procedure in **Synthetic procedures and characterization** in the supporting info). Subsequently, **15** was characterized using 2D NMR techniques (**Figure S112–S114**) and UV-Vis absorption spectroscopy (**Figure S116**). Its interaction with Fe²⁺ was investigated using ¹H NMR titration (**Figure S115**) revealing a low association constant (<10⁵ M⁻¹). Additionally, absorption titration was performed (**Figure S117**), which could be used to determine association constants K_{a1} and K_{a2} for a 1 : 2 complex (**15***2Fe²⁺). The program "BindFit" was then used to calculate $K_{a1} = 806 (\pm 13) \text{ M}^{-1}$ and $K_{a2} = 1789 (\pm 67) \text{ M}^{-1}$ (see newly introduced references 44–45 for further information).